

# Measurement report: Inland ship emissions and their contribution to NOₓ and ultrafine particle concentrations at the Rhine

Philipp Eger[1]*, Theresa Mathes[1,a]*, Alex Zavarsky[1], and Lars Duester[1]

[1]Federal Institute of Hydrology, Qualitative Hydrology, Koblenz, Germany
[a]now at: Environmental Chemistry and Air Research, Technische Universität Berlin, Berlin, Germany

*These authors contributed equally to this work.

*Correspondence to*: Philipp Eger (eger@bafg.de), Theresa Mathes (mathes@tu-berlin.de)

**Abstract.**

Emission plumes of around 4700 ship passages were detected between March 2021 and June 2022 in the Upper Rhine valley in Worms, Germany. In combination with ship-related data recorded via the Automatic Identification System (AIS), the plume composition of individuals ships was analysed and it was possible to quantify their contribution to the overall emission load. To obtain an integral picture of inland ship emissions, nitrogen oxides ($NO_x = NO + NO_2$) and carbon dioxide ($CO_2$) measurements in the gas-phase were combined with detailed particle-phase measurements including particle number concentration (PNC), particle size distribution (PSD) from 5 nm to 10 µm, particulate matter ($PM_1$ and $PM_{2.5}$), ultrafine particle fraction (UFP, diameter < 100 nm) and aerosol black carbon (BC). One measuring station was located in a bridge directly above the navigation channel and was especially helpful in deriving emission factors under real-world driving conditions for the fleet on the Upper Rhine. The other station was situated on a river bank at about 40 m distance to the shipping lane and was thus representative for the exposure of people working or living close to the Rhine. Inland ships contributed 1.2 µg m⁻³ or 7 % on average to the local nitrogen dioxide ($NO_2$) concentration at the bridge above the shipping lane. $NO_x$ concentrations were increased by 10.5 µg m⁻³ (50 %), PNC by 800 particles cm⁻³ (10 %), $PM_1$ by 0.4 µg m⁻³ (4 %) and BC by 0.15 µg m⁻³ (15 %). On the river bank a $NO_x$ increase of 1.6 µg m⁻³ (8 %) and an $NO_2$ increase of 0.4 µg m⁻³ (3 %) were observed. More than 75 % of emitted particles were found in the UFP range with a geometric mean particle diameter of 52±23 nm. Calculated emission factors (25–75 percentiles) were 26–44 g per kg of fuel for $NO_x$, 1.9–3.2 g kg⁻¹ for $NO_2$, 0.3–0.7 g kg⁻¹ for BC, 0.9–2.3 g kg⁻¹ for $PM_1$ and (1–3) × 10¹⁵ particles kg⁻¹ for PNC, with a large variability observed from ship to ship. Relating these values to ship-specific parameters revealed the importance of engine characteristics, i.e. vessels using old motors with low revolutions per minute (RPM) caused comparably high emission factors for both $NO_x$ and PNC. Comparison with emission regulation limits set by the Central Commission for the Navigation of the Rhine (CCNR) and the European Union (EU) showed that mean energy-dependent emission factors under real-driving conditions were slightly above the respective requirements based on controlled laboratory conditions. The results from this study underline the importance of



long-term measurements with high temporal resolution to reliably estimate the contribution of inland shipping to air pollution in cities along heavy traffic waterways and to monitor a potential future emission reduction when modernizing the fleet.

## 1 Introduction

Shipping is a potential contributor to air pollution in coastal regions, harbors and along inland waterways with high traffic density (Eyring et al., 2010; Viana et al., 2014; Karl et al., 2019; Tang et al., 2020; Fink et al., 2023). In Western Europe, where a large amount of goods is transported on rivers and canals, inland ships can increase nitrogen oxides ($NO_x = NO + NO_2$), particulate matter (PM), aerosol black carbon (BC) and ultrafine particle (UFP) concentrations on a local scale (Uherek et al., 2010; Van der Gon and Hulskotte, 2010; Keuken et al., 2014; Kurtenbach et al., 2016; Pohl et al., 2017; Krause et al., 2022). In particular the Rhine as Europe's most important waterway in terms of transport volume is of major significance as it connects densely populated cities where people are often exposed to air pollution from road transport, industry and residential heating (Uherek et al., 2010; WHO, 2021). In Germany around 50 billion tonne-kilometre (tkm) goods per year are transported by inland vessels, which makes it the EU country with the highest transport performance in this sector, closely followed by the Netherlands (CCNR, 2021). Commonly inland navigation is considered the most environmentally friendly mode of transport in terms of the volume of freight moved (Federal Ministry for Digital and Transport, 2019) and that is why the European Commission aims on strengthening the role of inland waterway transport and increasing the volume of goods transported on waterways by 50 % (European Commission, 2021). This consideration highlights the urgent need to better understand and therefore, more extensively monitor, the impact of ship emissions on air quality.

In recent years pollution from road traffic, sea-going ships and the negative impact of particulate matter on human health have become the subject of a vital scientific discussion (WHO, 2021; Sokhi et al., 2022), but there are only a few measurement studies dealing with European inland shipping in particular (van der Zee et al., 2012; Kurtenbach et al., 2016; Pohl et al., 2017; Kattner, 2019; Krause et al., 2022). Most of them took place at a fixed location for a short period of time or only concentrated on one specific compound. The only long-term study so far has been conducted in Duisburg at the Lower Rhine within the scope of the European research project "Clean Inland Shipping" (CLINSH) (Krause et al., 2022) where emission factors for $NO_x$ were reported and compared to on-board measurements directly at the exhaust line. Particulate matter and especially UFP from inland vessels have solely been investigated in more detail by Pohl et al. (2017), although the World Health organization (WHO) recommends the measurement of this particle size range and its integration into existing air quality monitoring (WHO, 2021). The difficulty here is that emitted UFPs are variable in their size and number concentration and can be affected by nucleation, condensation and coagulation processes, resulting in a strong dependence on wind conditions, plume age and distance between emitter and measurement location (Celik et al., 2020). Apart from the limited number of onshore and a few onboard (Schweighofer and Blaauw, 2009; Pillot et al., 2016) measurements, several modeling studies and emission inventories are available (Van der Gon and Hulskotte, 2010; Knörr et al., 2013; Korsten, 2018) which are, however, subject to uncertainties arising from inevitable assumptions to be made and from the use of generalized emission factors.





The impact of sea-going ships (and non-European inland ships) on air quality in coastal and port areas was more frequently investigated (Moldanová et al., 2009; Jonsson et al., 2011; Diesch et al., 2013; Alföldy et al., 2013; Beecken et al., 2014; Pirjola et al., 2014; Beecken et al., 2015; Seyler et al., 2017; Cao et al., 2018; Celik et al., 2020; Jiang et al., 2021; Kuittinen et al., 2021; Schwarzkopf et al., 2022), but due to different fuel types (especially the higher sulfur content) and ship dimensions the results are not transferable to inland ships. Permanent measurement stations that monitor air quality in the EU are usually located in cities near busy roads with a focus on road traffic. Although these stations are useful to check compliance with immission regulations for $NO_2$, $PM_{2.5}$ and $PM_{10}$ (European Union, 2008), they do not provide a deeper insight into the share of different emission sources (including road traffic, industry, agriculture, residential heating and shipping). In particular the distinction between emissions from inland vessels and diesel-powered vehicles is difficult as they use the same fuel (apart from colorants and additives), for which a maximum sulfur content of 0.001 % is allowed in Germany (10. BImSchV., 2010). In contrast to road traffic, inland vessels are generally older and emission regulations (especially for $NO_x$ and PM) are only valid for newly registered motors from 2003 on (CCNR (2022), overview given in Table S1 in the supplement). With the introduction of Euro V stage in 2020 (European Union, 2016) emissions limits for $NO_x$ and PM have been tightened but due to grandfathering and long motor lifecycles, the initiated transition to a low-emission inland fleet is slow.

Despite the growing relevance of inland shipping and because of the difficulty to use the above-mentioned existing monitoring networks, there is a critical lack of emission measurements for inland ships under real-driving conditions. We show that from measured ship plumes and position data from Automatic Identification System (AIS), emission factors (relative to the amount of fuel used) for several pollutants as well as the fractional contribution from ship traffic can reliably be derived. The presented dataset combines gas-phase ($NO_x$ and $CO_2$) with particle-phase (PM, BC, UFP and particle size distribution) measurements and delivers an integrative overview on the impact of inland vessels on air quality at the river Rhine. The wide measured particle size range of 5 nm to 10 µm, the high temporal resolution of ~1 s and a measurement period of more than one year provide a novel insight into particulate emissions from inland shipping. Emission factors representative for the fleet operating on the Upper Rhine section between Mannheim and Mainz and their dependence on ship parameters (e.g. motor power and RPM) were derived from the dataset and offer the potential for the enhancement of emission inventories and transport models.

## 2 Methods

### 2.1 Measurement site

Measurements were carried out at two stations along the Upper Rhine (Rhine kilometer 444) in Worms, Germany, over a period of more than one year (March 2021 to August 2021 and November 2021 to June 2022). The city of Worms is located on the westside of the river Rhine, whereas on the eastside the landscape is dominated by rural structures including agricultural areas, as indicated in Fig. 1a. Meteorological conditions varied throughout the year but were mainly characterized by northern and south-western winds (Fig. 1b).





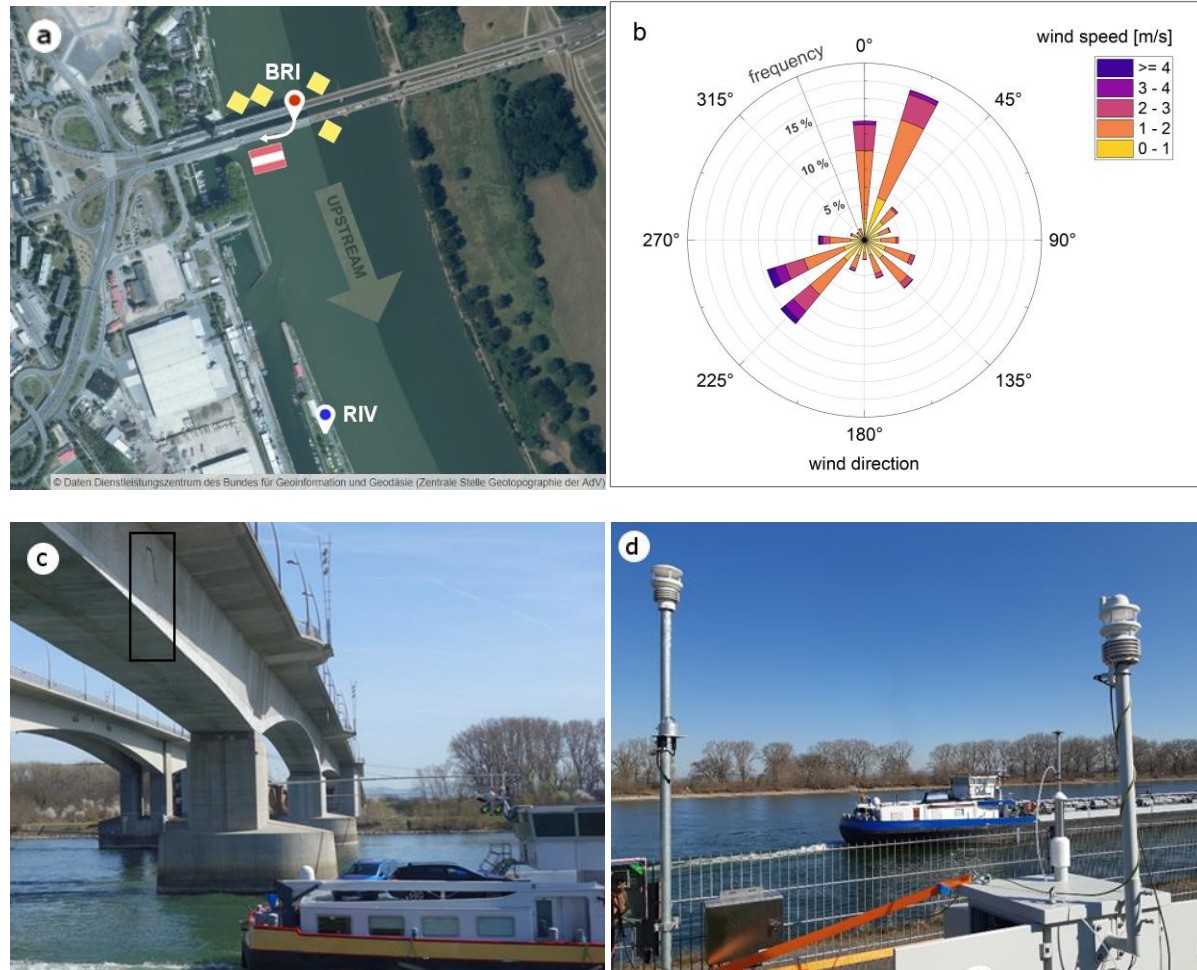

**Figure 1:** (a) Satellite image with measurement station at the bridge (BRI) and station at the river bank (RIV). Data: © GeoBasis-DE / BKG (2023). Ships travelling upstream preferably used the western shipping lane (yellow diamond means passage allowed, two yellow diamonds mean passage recommended) whereas ships travelling downstream were forced to the eastern shipping lane (red-white-red flag means passage forbidden). (b) Wind rose for the whole measurement period. (c) Setup at BRI with sampling line framed in black. (d) Setup at RIV.

One station was located in the "Nibelungen" bridge (labelled BRI, Fig. 1c) at a mean height of ~ 10 m above the shipping lane (depending on water level) where ships were travelling upstream (further referred to as upstream ships), at a distance of 40 m to the western river bank. The instruments were located in a room inside the bridge which provided protection from rain, high humidity and rapid temperature fluctuations. Instrument-specific sampling lines of 4–5 m length were fed through a small hole in the wall and led downwards to a point sharply below the edge of the bridge's base to enable an undisturbed incoming flow. For a period of two month, the sampling inlet was also moved to the lane where ships were travelling downstream (downstream ships) at 100 m distance to the western river bank. The other station was located about 500 m upstream from the bridge on the



river bank (labelled RIV, Fig. 1d) at a distance of 40 m respectively 100 m to the shipping lanes. Here the instruments were placed inside an air-conditioned container (25 °C), with sampling lines of approximately 1 m length for the gas-phase measurements and a customized sampling head for the particle-phase measurements.

The two stations were chosen to be as close as possible to the shipping lane to enable plume identification but at sufficient
distance to the city center to avoid strong interferences from road traffic. BRI was primarily set up to derive emission factors with a minimum uncertainty since the point of measurement is very close to the emission source. RIV was installed in February 2022 to monitor pollutant immissions at a more representative location in order to compare them with local background levels. At RIV the signal was strongly dependent on the predominant wind direction with noticeable signals found for north-easterly winds whereas at BRI ship plumes were detected mostly independently from wind conditions, although with varying
magnitude.

### 2.2 Instruments

Table 1 lists the instruments used to measure $CO_2$, $NO_x$, particle number concentration (PNC), particle size distribution (PSD), particulate matter (PM) concentration and black carbon (BC). In order to capture the shape of the ship plumes in real time, all instruments were operated at the highest possible temporal resolution (1 to 6 s).

### 2.2.1 Trace gas measurements

$CO_2$ was measured by non-dispersive infrared spectroscopy (NDIR) with a LI-850 device (LI-COR Biosciences) and a temporal resolution of 1 s. For the combined measurement of $NO_2$ and $NO_x$ an ICAD instrument (Airyx) based on the iterative cavity-enhanced DOAS method (Horbanski et al., 2019) was applied. It simultaneously measured $NO_2$ and $NO_x$ at a high temporal resolution of ~2 s and contained an integrated $CO_2$ sensor (FlowEvo) serving as an additional $CO_2$ measurement
(correlation with Licor device: slope = 0.94, $R^2$ = 0.97, see Fig. S1). Parallel $NO_x$ measurements at RIV were carried out with the $NO_x$ Analyzer Model 405 (2B Technologies, 5 s temporal resolution) which directly measures $NO_2$ via absorption at 405 nm and NO by converting it to $NO_2$ via gas-phase titration (Birks et al., 2018). $O_3$ was monitored via absorption spectroscopy with the ozone monitor Model 205 (2B Technologies). For the gas-phase instruments we used perfluoroalkoxy (PFA) tubing with 4 mm inner diameter (ID) and 6.35 mm outer diameter (OD). The residence time in the sampling lines was in the order
of 1–2 s so that no significant losses to surfaces are expected for the gases measured in this study. The instrument performance was checked on a regular basis (zero and span tests). For NO, $NO_2$ and $O_3$ this was done via the Calibrator Model 714 (2B Technologies) with a user-adjustable output of the three gases and an $NO_2$ permeation source. For $CO_2$ a gas bottle with a mixing ratio of 2000 ppm and a reservoir of soda lime as a zero-air reference were used.

### 2.2.2 Particle measurements

PNC and PSD from 6 to 520 nm diameter were detected in 16 size channels by a fast (electric) mobility particle sizer spectrometer (TSI, FMPS 3091) with a frequency of 1 Hz. Particles ranging from 0.25 to 2.5 µm diameter were captured



optically by an aerosol spectrometer (Grimm, 11-D) with a time resolution of 1 s. During the first weeks the size range was extended to 32 µm diameter (in 6 s mode) but no significant increase in coarse mode particles > 2.5 µm was observed in ship plumes at BRI. This finding was confirmed by parallel measurements with a similar optical particle counter (Grimm, EDM 180) at RIV. Consequently, further measurements with the 11-D were only carried out up to 2.5 µm diameter to gain a higher

5    temporal resolution.

**Table 1:** Instruments applied at BRI and RIV and their specifications.

| Instrument | Variables | Time res. | Precision[a] (1σ) | Accuracy[b] | Size range | BRI[c] | RIV[c] |
|---|---|---|---|---|---|---|---|
| LI-850 (LI-COR) | $CO_2$ | 1 s | 0.3 ppm | 1.5 % | - | x | x |
| ICAD (Airyx) | $NO_2$, $NO_x$ $CO_2$ | 2 s | 0.4 ppb 1.0 ppm | | - | x | (x) |
| Model 405 (2B) | $NO_2$, $NO_x$ | 5 s | 4.0 ppb | 2.0 % | - | | x |
| Model 205 (2B) | $O_3$ | 2 s | 1.0 ppb | 2.0 % | - | x | |
| FMPS 3091 (TSI) | PNC, $PM_{0.1}$, $PM_{0.5}$, size distribution | 1 s | | | 6–520 nm | x | |
| 11-D (Grimm) | PNC, $PM_{0.1}$, $PM_{0.5}$, size distribution | 1 s (6 s) | | | 0.25–2.5 µm (0.25–32 µm) | x | |
| EDM 180 (Grimm) | PNC, $PM_{0.1}$, $PM_{0.5}$, size distribution | 6 s | | | 0.25–32 µm | | x |
| SMPS+C (Grimm) | Particle number, size distribution, | 85 s per scan | | | 5–350 nm | | x |
| AE33 (Magee) | BC | 1 s | 0.4 µg m$^{-3}$ | | 0–2.5 µm | x | |
| AIS receiver (Watcheye R) | Ship position data and characteristics | 1 s | | | - | x | x |
| Weather station[d] (Lufft) | T, RH, wind speed, direction, precipitation, global radiation | 10 min | | | - | | x |

[a]The precision is derived from noise in our measurements at a constant representative concentration and highest possible time resolution.
[b]Specified by the manufacturer. [c]Columns indicate at which station the instruments were operated: x = full time, (x) = partly. [d]Temperature
10    T, relative humidity RH.

For all particle measurement devices, particulate matter concentration ($PM_1$, $PM_{2.5}$) was derived from the PNC in the corresponding channels, assuming a constant density of 1.7 g cm$^{-3}$. This estimate is based on numerous literature studies on diesel exhaust and urban aerosols (Barone et al., 2011; Celik et al., 2020; Wang et al., 2021), also considering the relatively large BC fraction (~2 g cm$^{-3}$) measured in our study. Nevertheless, the number is subject to a substantial uncertainty as the

15    exact particle composition is unknown and also changing over time. From the measured PSD the mode diameter ($D_{mode}$) and





the geometric mean diameter (GMD) were determined. To measure BC an aethalometer (Magee Scientific, AE33) with a time resolution of 1 s was used. The BC mass concentration is derived from the measurement of absorbance at 880 nm and real-time compensation of the filter loading effect (Drinovec et al., 2015). The fractional contribution from biomass burning can be derived from the difference in absorption at 470 nm and 880 nm. The AE33 was operated with a 2.5 µm cyclone to be in

line with the optical particle measurement. For the particle-phase measurements 3 mm ID stainless steel tubing (Grimm 11-D) and conductive silicon tubing with 7.87 mm (FMPS) respectively 6.35 mm (AE33) ID were used. Transmission losses were calculated using the particle loss calculator tool (von der Weiden et al. (2009), see Fig. S2) and resulted in overall losses smaller than 10 % for all three instruments and particle diameters up to 2.5 µm. Comparison between the Grimm 11-D at BRI (long tubing) and the EDM 180 at RIV (dedicated inlet system) confirmed these small particle losses.

**2.2.3 Meteorological and AIS data**

A permanent weather station in the city center (~10 m above the ground, operated by the Landesamt für Umwelt Rheinland-Pfalz) at ~1 km distance to BRI provided meteorological data throughout the whole measurement period. From February 2022 on this was complemented by a weather sensor directly located at RIV (Lufft / OTT HydroMet, WS700-UMB), measuring temperature (T), relative humidity (RH), wind speed, wind direction, precipitation intensity and global radiation. In general,

both stations were in good agreement with respect to the prevailing wind direction (although wind speed and relative humidity were higher at the river banks).

Ship information and position data were recorded in a pseudonymized form using an AIS receiver (Watcheye R) and a program written in Python code. The Rhine in Worms is frequented by approximately 100 ships per day, which are mainly cargo (37 %) and tanker ships (35 %), but also a few passenger boats (4 %) and vessels of other type (not further specified or incorrectly

assigned). The speed of the vessel relative to the river current (further referred to as speed through water, STW) was derived from the speed over ground (SOG) recorded via AIS and the flow velocity of the Rhine via SOBEK model data of 1-day resolution (Hydrokontor, 2014) (see Fig. S3). On average, upstream ships travelled at a SOG of 2.8±0.4 m s$^{-1}$ and downstream ships at 4.8±0.7 m s$^{-1}$. Correcting for the velocity of the Rhine (~1.3 m s$^{-1}$ on average), STW yielded 4.1 and 3.6 m s$^{-1}$, i.e. upstream ships were operated at a slightly higher engine load. The loading status can theoretically be derived from the AIS

data as well but is not considered sufficiently reliable as it lies within the responsibility of the ship owner to keep the information up-to-date. The CCNR (2018) reports that vessels transporting mineral oil products (tankers) on the Upper Rhine are on average more heavily loaded when travelling upstream whereas for sand and construction materials (cargo) the situation is vice versa.

A database with additional technical vessel information for Rhine ships (mainly about the engine), was provided by the IVR

(International Association for the representation of the mutual interests of the inland shipping and the insurance and for keeping the register of inland vessels in Europe) and processed in a pseudonymized form to comply with data protection guidelines. For the fleet operating on the Upper Rhine in Worms we calculated a median age of the ships' main engine of 12 years (25–



75 percentiles: 4–17 years) but the result might potentially be biased by the fact that newer motors are more likely to be listed in the database than older ones.

## 2.3 Data analysis

### 2.3.1 Peak identification and ship assignment

Data from gas and particle phase instruments were combined with the AIS dataset to identify potential peaks from passing ships. From the AIS position data (transmitted every 10 s) the point in time with the smallest distance to the point of measurement was determined and further used as a reference ($t_0$). In the next step the atmospheric background was calculated for each measured parameter using a low-pass filtered time series in form of a running median with 10 min window size. By subtracting this value from the raw signal (see Fig. 2), most of the background variability (stemming from chemical and

meteorological processes in the atmosphere and non-ship emission sources) was removed from the dataset and only distinct peaks with a short duration of typically several tens of seconds remained. In the second step an algorithm was applied to automatically identify potential ship peaks by defining several criteria to be fulfilled, illustrated in Fig. 2.

(i)     A minimum amount of 3 consecutive data points above a pre-defined threshold, which is defined as 4 times the standard deviation ($\sigma$) of the signal in a 30 s window before the potential peak start ($t_{start}$).

(ii)    A minimum peak height to avoid that atmospheric variability or noise is interpreted as a ship peak.

(iii)   Peaks occurring in periods with high atmospheric variability were skipped when a precise separation from the background was impossible.

(iv)    The peak end ($t_{end}$) was defined as the point in time after which the signal decreased below the threshold level for more than 20 s.

(v)     The peak duration was limited to $\tau = 240$ s since longer peaks indicated an unusual maneuver (e.g. a turn).

Descriptive examples and further information can be found in the SI (Table S2, S3 and Fig. S4). If all criteria were fulfilled, peak duration, peak maximum, cumulative peak area and time of occurrence were stored. By using the peak area instead of the maximum value, biases from different measurement frequencies of the instruments were avoided. In the second step the peaks were assigned to passing ships if the following criteria were met:

(vi)    The peak was observed within the interval $\Delta t =$ -30 to +120 s relative to the ship passage that is defined via the AIS position closest to the measurement station (expanded to -60 to +240 s at RIV).

(vii)   If more than one ship was present within a time span of 120 s (240 s at RIV), the corresponding peaks were rejected from the analysis, as a clear attribution to one single ship was impossible.

The remaining peaks provided the basis for the calculation of specific emission factors and the additional contribution

from inland shipping.



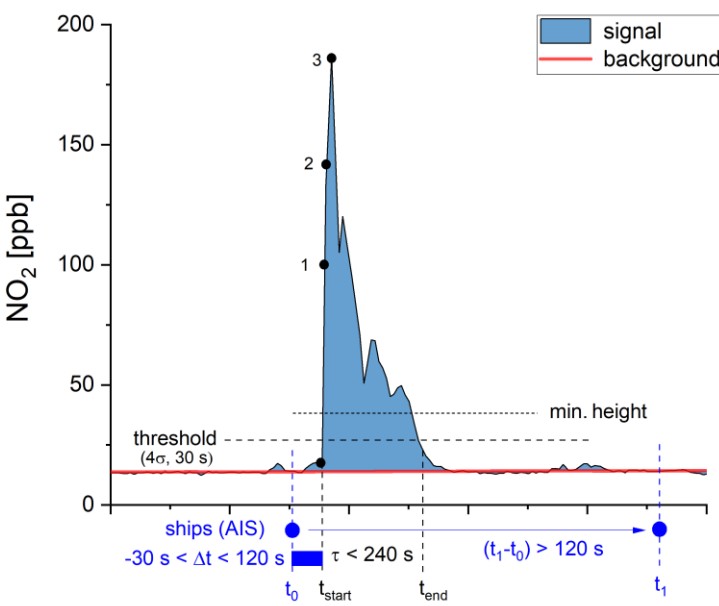

**Figure 2:** Exemplary peak illustrating the criteria applied for peak identification and ship assignment explained in Sect. 2.3.1.

### 2.3.2 Additional contribution from shipping

The average contribution from inland shipping to the overall emission load was calculated from the sum of all peaks that could
be clearly attributed to a single ship passage (see criteria in 2.3.1) relative to the observed background value within a particular
measurement period. This approach represents a lower limit estimate, since a non-negligible number of peaks is excluded from
the analysis (e.g. periods with too high ship traffic density or increased atmospheric variability), but prevents the results from
being biased by peak-like-structures from non-ship sources.

### 2.3.3 Emission factors

For each peak that was successfully assigned to a ship passage the emission factor $E_X$ of a pollutant X (expressed in g of X per
kg of fuel used) was derived from Eq. (1a). Here $C_X$ is the peak area of pollutant X, $C_{CO2}$ the peak area of $CO_2$ (expressed in
ppb s respectively ppm s), $M_X$ the molar mass of X (46 g mol⁻¹ for $NO_2$ and $NO_x$) and $M_{CO2}$ = 44 g mol⁻¹ the molar mass of
$CO_2$. Furthermore, the amount of $CO_2$ emitted per kg of burned fuel is estimated as $f_1$ = 3150 g kg⁻¹, assuming a carbon content
in diesel fuel of 86 weight percent (wt %) C and complete combustion to $CO_2$ (Cooper, 2001; Kurtenbach et al., 2016). For
particulate matter (PM) and number concentration (PNC) the emission factors were derived from Eq. (1b) and (1c), with $C_{PM}$
being the peak area of PM (expressed in µg m⁻³ s), $C_{PNC}$ the peak area of PNC (expressed in particles cm⁻³ s) and
$V_{air}$ = 24.45 L mol⁻¹ the molar volume of air (at 25 °C and 1013 hPa).

$$E_X \text{ [g kg⁻¹]} = f_1 \times \left(\frac{C_X}{C_{CO2}}\right) \times \left(\frac{M_X}{M_{CO2}}\right) \qquad\qquad \text{(Eq. 1a)}$$



$$E_{PM} \, [\text{g kg}^{-1}] = f_1 \times \left(\frac{C_{PM}}{C_{CO2}}\right) \times \left(\frac{V_{air}}{M_{CO2}}\right) \qquad \text{(Eq. 1b)}$$

$$E_{PNC} \, [\text{particles kg}^{-1}] = f_1 \times \left(\frac{C_{PNC}}{C_{CO2}}\right) \times \left(\frac{V_{air}}{M_{CO2}}\right) \qquad \text{(Eq. 1c)}$$

For Eq. (1a-c) to be valid we assume that pollutant X and $CO_2$ are diluted in a similar manner during transport from the point of emission to the point of measurement. This is a reasonable assumption for most components ($NO_x$, BC, PNC) as the plume age is in the order of seconds to a few minutes and mixing ratios will be conserved. The only exception is NO which is rapidly converted to $NO_2$ in the presence of $O_3$. Thus, the results for $NO_2$ were corrected for the amount of $NO_2$ that has been produced during plume transport. To calculate the initial concentration at the time of emission, the amount of $O_3$ that has been consumed by the reaction with NO was subtracted from the measured $NO_2$ signal, resulting in a somewhat lower concentration of $NO_2$ at the time of emission (while $NO_x$ is conserved, Eq. 2). The initial $NO_2$-to-$NO_x$ ratio is then defined by Eq. (3).

$$\Delta(NO_2)_{initial} = \Delta(NO_2)_{measured} - \Delta(O_3)_{decrease} \qquad \text{(Eq. 2)}$$

$$\left(\frac{NO2}{NOx}\right)_{initial} = \frac{\Delta NO2_{initial}}{\Delta NOx_{measured}} \qquad \text{(Eq. 3)}$$

Emission factors were calculated separately for each measured component. As part of the quality check, emission factors were only considered reliable if for $CO_2$ and the analyte of interest the peak duration did not differ by more than a factor 3 and the time of occurrence did not deviate by more than 20 s. The application of these quality control criteria helped in minimizing outliers and systematic uncertainties in the dataset. An exemplary plausibility check was carried out for data from December 2021 (~500 peaks), i.e. peaks considered unreliable (< 4 %) were manually removed from the dataset. The analysis revealed that both mean emission factors and the mean contribution from shipping hardly changed (< 2 %) when applying the additional manual filtering procedure. In addition, missing peaks in BC were checked in detail to decide whether the peak was just not automatically recognized or if there was really no BC present (e.g., due to applied particle filtering technology). In the latter case, which was true for ~ 5 % of all ship plumes, the BC peak height was manually set to zero, marginally reducing the calculated mean emission factor.

For comparison with emission regulations the emission factor in g per kg was transformed into an energy-related emission factor (expressed in g per kWh) using Eq. (4) and a specific fuel consumption reported for inland ships of $f_2 = 220$ g $(\text{kWh})^{-1}$ (Federal Ministry for Digital and Transport, 2020), which is based on an analysis of inland vessel engine test-bed reports within a performance range of 25 % to 100 % of nominal power.

$$E_X' \, [\text{g kWh}^{-1}] = E_X \, [\text{g kg}^{-1}] \times f_2 \qquad \text{(Eq. 4)}$$

Although the fuel consumption in reality is not independent of driving conditions as the efficiency decreases at lower loads, the variability when travelling under typical operating conditions (like observed in Worms) is very small.

### 2.3.4 Dependence on ship parameters and cluster analysis

The division of vessels into ship classes according to their length and width followed a CEMT scheme (CEMT, 1992) adjusted by Krause et al. (2022). Individual ships were further classified with respect to length, width, engine power, RPM and motor



age using a *k*-means clustering algorithm. After (min-max) normalization of the data, clustering was performed separately for the $NO_x$ and particle dataset. A number of $k = 5$ clusters was found to provide the most reliable results.

## 3 Results and discussion

### 3.1 General overview

Emission plumes of inland ships consist of a mixture of gaseous and particulate compounds, from which $CO_2$, NO, $NO_2$, PNC, $PM_1$, $PM_{2.5}$ and BC were analyzed in this study. For the two measurement stations in Worms (BRI and RIV) a total number of 30 883 ship passages from 1781 different vessels were registered via AIS at BRI (10 703 passages from 1314 ships at RIV). Following the procedure described in Sect. 2.3.1, a fraction of ~13 % (4060 peaks from 1181 ships) could be clearly attributed to a single ship passage at BRI (680 peaks from 476 ships at RIV). This relatively low yield arises from meteorological

conditions and strict criteria set for the peak assignment algorithm. These criteria were necessary to avoid biases by false peak assignment but still provided a sufficiently large and representative sample. A one-day example for BRI is shown in Fig. 3 where ship plumes are clearly visible as distinct spikes of various magnitude and shape and a drop in $O_3$.

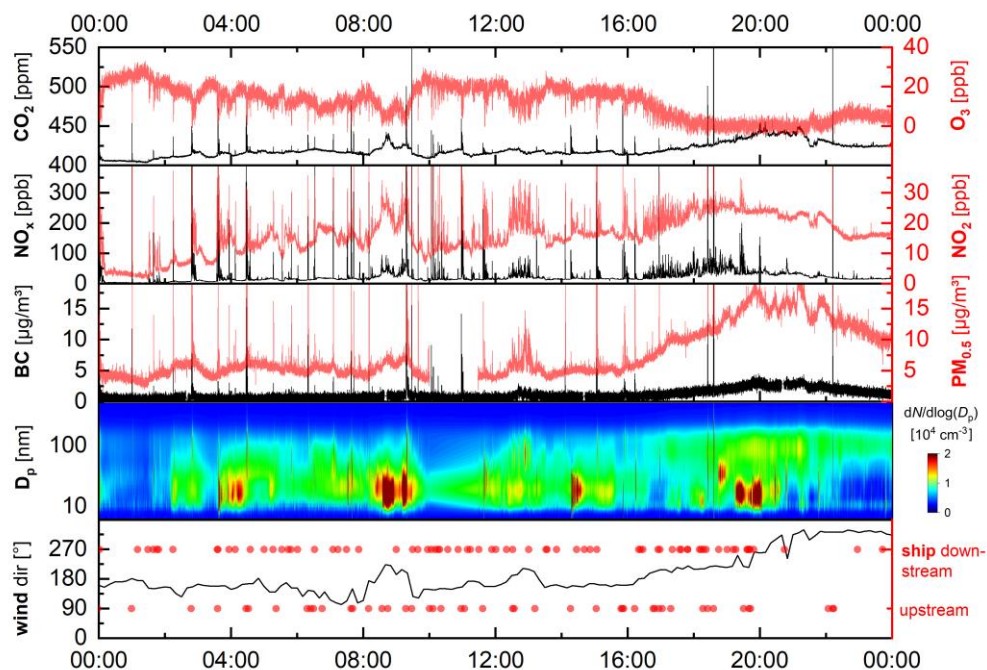

**Figure 3:** Exemplary time series for $CO_2$, $O_3$, $NO_x$, $NO_2$, BC, $PM_{0.5}$ (in this study similar to $PM_1$ and $PM_{2.5}$), $dN/dlog(D_p)$ with number concentration $N$ and particle diameter $D_p$, wind direction and ship travelling direction at BRI for 10 December 2021.



### 3.1.1 Nitrogen oxides (NO$_x$)

At BRI upstream ships caused median NO$_x$ and NO$_2$ peak heights of 1053 ppb respectively 91 ppb with observed maxima of 7851 ppb and 3444 ppb (based on $N_{NOx}$ = 2912 and $N_{NO2}$ = 3034 peaks). These relatively large short-term peak values can be explained by the choice of the inlet location situated directly above the shipping lane. For downstream ships the relative peak

height was a factor of 2–3 lower ($N_{NOx}$ = 288 and $N_{NO2}$ = 306) as they travelled with higher SOG resulting in a lower emission density along the track. At RIV median concentration peaks of 186 ppb NO$_x$ and 37 ppb NO$_2$ and maxima of 3480 and 160 ppb were registered ($N_{NOx}$ = 614 and $N_{NO2}$ = 531). As diesel engines mainly emit NO, a median NO$_2$-to-NO$_x$ ratio of 0.11 was observed at BRI. The initial ratio at the point of emission (see Sect. 2.3.3) was determined as 0.07, which is in good agreement with the ratio derived by Kurtenbach et al. (2016) for inland ships on the Lower Rhine. This value is also typical for diesel

engines without exhaust gas aftertreatment systems, assumed to constitute the largest portion of the inland fleet (see Sect. 2.2.3), whereas the use of catalysts and diesel particulate filters (DPF) usually results in a somewhat higher ratio of 0.25–0.30 (Kurtenbach et al., 2016). At RIV, emission plumes had more time to chemically react in the atmosphere before reaching the sampling inlet. As a consequence, more NO was converted to NO$_2$ and the NO$_2$-to-NO$_x$ ratio increased to 0.19. For further analysis only the initial NO$_2$ concentration was used, as the measured NO$_2$ in the plume was strongly dependent on the transport

time and not characteristic for a particular ship.

### 3.1.2 Particles (PNC, PSD, PM, BC)

The analysis of the PSD showed almost no particles in the coarse mode (2.5 to 10 µm diameter) present in ship plumes. The majority of emitted particles belonged to the ultrafine particle fraction (UFP, <100 nm) with a fractional contribution of 76±16 % relative to the total number of particles observed from 6 nm to 2.5 µm and 18±20 % relative to the total particle mass.

Since the FMPS (size range 6–520 nm) covered on average > 99.9 % of all particles detected in ship plumes, further results of PNC, PM$_1$ and PM$_{2.5}$ only refer to this instrument.

Like for NO$_x$, the peak height values of PNC for downstream vessels were on average 2–5 times lower than for upstream vessels with an observed mean of $(0.6±1.4) \times 10^6$ particles cm$^{-3}$ for PNC ($N_{PNC}$ = 1503) and 323±417 µg m$^{-3}$ for PM$_1$ ($N_{PM}$ = 1503). The mean peak area of upstream vessels amounted 2710±3192 µg m$^{-3}$ s with a mean peak duration of 72±52 s.

A similar picture was obtained for BC, where peak height values of upstream vessels of 54±65 µg m$^{-3}$ ($N_{BC}$ = 1620) were around 5 times higher than for downstream vessels. These results are consistent with a measurement study by Pohl et al. (2017) at the Upper Rhine. The high variability in our dataset can be explained by differences in meteorological conditions, ship position relative to the measurement station and vessel characteristics, and is also in line with other studies (Keuken et al., 2014; Pohl et al., 2017). Peak heights of inland ships measured in direct proximity to the fairway were of similar magnitude

than emissions from sea-going ships at a larger distance of about 200-800 nm (Pirjola et al., 2014).



## 3.2 Additional contribution from shipping

The additional contribution from inland shipping to local background concentrations (composed of emissions from road traffic, industry, agriculture and residential heating) was calculated for both measurement stations in Worms as described in the methods section. For each analyte a non-negligible background was reliably quantified and subtracted from significantly higher

ship peaks, detected over a long period of time. The results are shown in Table 2 and, as explained in the method section, these values represent lower limits.

**Table 2:** Monthly averaged contribution from shipping at both measurement stations at the Rhine in Worms.

| Analyte | Monthly additional contribution at BRI | $N_{peaks}$[a] | Relative contribution | Monthly additional contribution at RIV | $N_{peaks}$ | Relative contribution |
|---|---|---|---|---|---|---|
| $CO_2$ | 0.7–1.4 mg m$^{-3}$ | 3204 | 0.1–0.2 % | 0.07–0.13 mg m$^{-3}$ | 1083 | ~0.01 % |
| $NO_x$ | 6.9–14.3 µg m$^{-3}$ | 4013 | 24–68 % | 1.0–2.2 µg m$^{-3}$ | 1806 | 6–11 % |
| $NO_2$ | 0.7–1.6 µg m$^{-3}$ | 4607 | 3–10 % | 0.3–0.5 µg m$^{-3}$ | 2374 | 2–4 % |
| BC | 0.1–0.2 µg m$^{-3}$ | 3160 | 7–30 % | 0.01–0.02[b] µg m$^{-3}$ | - | - |
| PNC | 270–1540 particles cm$^{-3}$ | 4262 | 3–16 % | 27–154[b] particles cm$^{-3}$ | - | 1–3 % |
| $PNC_{0.1}$ (UFP) | 240–1390 particles cm$^{-3}$ | 4262 | 3–18 % | 27–139[b] particles cm | - | 1–3 % |
| $PM_1$ (= $PM_{2.5}$) | 0.1–0.7 µg m$^{-3}$ | 4262 | 1–6 % | 0.01–0.07[b] µg m$^{-3}$ | - | ~1 % |

[a]$N_{peaks}$ is the total number of peaks the calculation was based on. [b]Values at RIV are estimated from values at BRI, assuming the same

dilution factor than for $CO_2$.

Monthly average concentrations of $NO_x$ were increased by 6.9–14.3 µg m$^{-3}$ at BRI (above the shipping lane) and by 1.0–2.2 µg m$^{-3}$ at RIV (river bank). This corresponds to an additional burden of 24–68 % respectively 6–11 %, relative to extrapolated background levels without the shipping contribution. The additional $NO_2$ load was 0.7–1.6 µg m$^{-3}$ (3–10 %) at BRI and 0.3–

0.5 µg m$^{-3}$ (2–4 %) at RIV, whereby the higher dilution factor at RIV was partly compensated by the increased NO to $NO_2$ conversion. Although a mean $NO_2$ contribution of a few percent can be considered relatively small, we note that it is strongly depending on the meteorological conditions and can be substantially higher for a short period of time (e.g. up to 20 % on 16 April 2022 with wind from the eastern sector). In the case of RIV the derived values are in particular representable for the exposure of people living or working very close to the Rhine. Additional loads at BRI should more be considered as an upper

limit for a maximum potential exposure. The increase in $CO_2$ due to shipping was ~0.1 % at BRI and ~0.01 % at RIV and thus negligible in terms of air quality (although relevant as a greenhouse gas contribution).

For BC the contribution at BRI was 0.1–0.2 µg m$^{-3}$ corresponding to 7–30 %, whereas $PM_1$ was increased by 0.1–0.7 µg m³ respectively 1–6 %. The increase of 3–16 % in PNC was dominated by small particles in the UFP range. Here our results underline the need for integrating UFP and BC measurements in air quality monitoring, as they are emitted by diesel engines

in significant quantities and the fraction is highly relevant with respect to human exposure (WHO, 2021). Since particles were





not measured at RIV with a sufficiently high temporal resolution to analyze the corresponding peaks (see Table 1), we estimated these values from the additional contribution derived at BRI by assuming the same dilution factor as for $CO_2$, i.e. a factor of ~10. Here we assumed that particle coagulation and chemical surface reactions were negligible due to the short atmospheric residence time of several seconds to a few minutes. This is in accordance with Kuittinen et al. (2021) who observed

only marginal changes in the mean particle diameter over several minutes after emission. The background PNC required to calculate the relative contribution was taken from the (low temporal resolution) SMPS measurement.

### 3.3 Emission factors

Emission factors $E_X$ (in g per kg fuel) were calculated for all measured components via Eq. (1a–c) (see Sect. 2.3.3). The analysis was performed for station BRI only to minimize uncertainties arising from the smaller sample size and the larger

distance between ship and measurement inlet at RIV. Results are illustrated as box plots in Fig. 4 (for detailed statistics see Table S4).

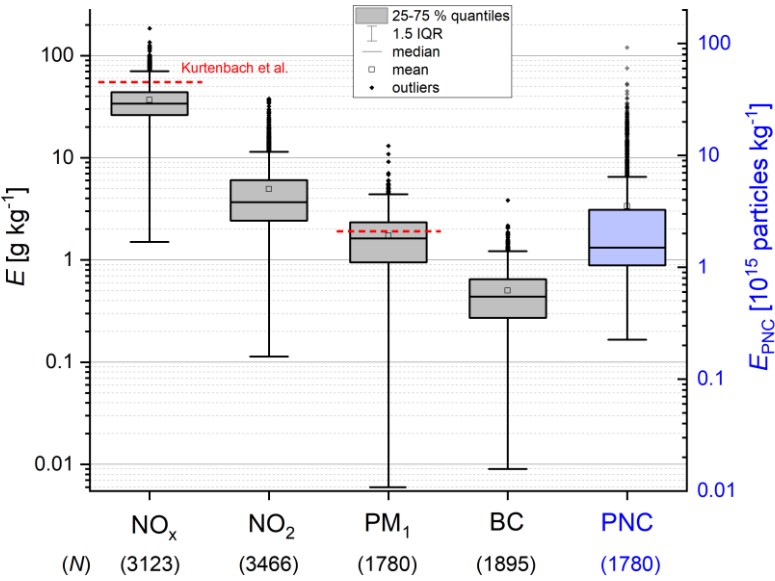

**Figure 4:** Boxplot of emission factors ($E$) measured at BRI and comparison with values reported by Kurtenbach et al. (2016) for the Lower Rhine in red. IQR = inter-quartile range. Numbers in parenthesis ($N$) indicate the sample size.

We observed a large variability of emission factors for every component. This can be traced back to differing operating conditions and ship parameters, including high emitters as well as modern ships with exhaust gas aftertreatment and will be discussed in Sect. 3.4 and 3.5. For $NO_x$ a mean value of $E_{NOx} = 37\pm16$ g kg$^{-1}$ was derived for the Upper Rhine section in Worms. This is significantly lower than the $52\pm3$ g kg$^{-1}$ reported by Kurtenbach et al. (2016) for the Lower Rhine between Wesel and Emmerich. For the Lower Rhine in Wedel a value of $40.9\pm27.6$ is reported by Kattner (2019) but here only inland



ships with a length < 100 m were included in the analysis. The differences might be explained by different fleet characteristics and operating conditions. For further investigation we classified the ships by their length and width following the modified CEMT scheme suggested by Krause et al. (2022). For the most common ship classes observed in our study the differences in mean values were less than 15 % and the variability within single ship classes was high (see Table S5 for details) so that no clear trend could be derived. Compared with emission factors from the study of Krause et al. (2002) for the Lower Rhine close to Duisburg harbor (converted from g s$^{-1}$ to g kg$^{-1}$ using the fuel consumption scenario suggested by the authors), our values are on average 20 to 40 % lower. This again indicates potential differences in fleet composition, load status and operating conditions between Lower and Upper Rhine. In general, NO$_x$ emission factors derived for inland vessels are lower than the range of 53–68 g kg$^{-1}$ observed for seagoing vessels (Diesch et al., 2013; Beecken et al., 2014; Kattner, 2019). The NO$_2$ emission factor of 2.9±2.5 g kg$^{-1}$ (Fig.4) at BRI was an order of magnitude lower than for NO$_x$, as during combustion most of the NO$_x$ is emitted as NO. It has to be kept in mind that the emitted NO is rapidly converted to NO$_2$ so that the NO$_2$-to-NO$_x$ ratio increases during plume transport. For this reason, NO$_2$ emission factors were not explicitly reported in other studies.

The emission factor for PNC was $E_{PNC} = (3.5±5.5) \times 10^{15}$ particles kg$^{-1}$ with the majority of particles situated in the UFP range (> 90 %). For the mass concentration we obtained $E_{PM} = 1.7±1.1$ g kg$^{-1}$ with about 30 % attributed to aerosol BC. For PM$_1$ a lower limit of 1.9 g kg$^{-1}$ was reported by Kurtenbach et al. (2016) which is in line with our measurements, although their data was generated by an OPC detecting only particles above 0.25 µm diameter. According to our results they may underestimate the real-world PM emission factor. Further comparison with literature data is unfortunately impossible, as from the best of the authors' knowledge no other studies reporting PM or PNC emission factors for inland ships are available. For sea-going vessels $E_{PM}$ values ranging from 2 to 3.3 g kg$^{-1}$ are very similar to our results, whereas $E_{PNC}$ was 2 to 7 times larger (Petzold et al., 2008; Jonsson et al., 2011; Lack and Corbett, 2012; Diesch et al., 2013; Beecken et al., 2014; Celik et al., 2020), probably due to the different fuel composition including the before mentioned higher sulfur content. The BC emission factor derived in this study ($E_{BC} = 0.5±0.3$ g kg$^{-1}$) is in the upper range of values reported for sea-going ships (Petzold et al., 2008; Diesch et al., 2013; Celik et al., 2020).

### 3.3.1 Comparison with emission regulation limits

Calculated emission factors in g kg$^{-1}$ were converted to energy-related values expressed in g kWh$^{-1}$ following Eq. (4) and compared with existing emission regulations (see Table S1). This includes limits set by the Central Commission for the Navigation of the Rhine (CCNR, 2022) for newly registered motors since 2003 (CCNR I) respectively 2007 (CCNR II), as well as the Euro IIIa standard (which for the majority of inland ships is similar to CCNR II) and the recently introduced Euro V stage (2020), proposed by the EU (European Union, 2016). Here it is important to point out that these emission limits are based on test bench measurements under controlled laboratory conditions, whereas our results represent emission factors under real-world driving conditions for this particular Rhine section. Consequently, a limit exceedance cannot directly be concluded from our measurement data. Nevertheless, derived emission factors provide valuable insights since vessels at the Rhine in Worms are operated under typical driving conditions and thus a reasonable agreement should be achieved. For NO$_x$ the





resulting value of (8.1±3.5) g (kWh)$^{-1}$ is approximately in line with emission limits for new motors since 2003 (CCNR I, see Table S1). For PM a mean emission factor of (0.38±0.24) g (kWh)$^{-1}$ was derived in our study, being slightly below CCNR I and above CCNR II standard. For PNC a limit of $1 \times 10^{12}$ particles (kWh)$^{-1}$ was for the first time introduced with Euro V. Our calculated mean value of $(0.8±1.2) \times 10^{15}$ particles (kWh)$^{-1}$ exceeds this limit by 3 orders of magnitude but this is not surprising

since up to now very few ships have been equipped with a new motor and such low values can only be achieved with the help of the DPF technology. A direct comparison for PM and PNC is difficult, anyway, as the emission limit refers to solid particles with diameter > 23 nm, whereas our measurements include both the solid and the semi-volatile fraction of particles with diameter > 6 nm. According to Petzold et al. (2008) and Giechaskiel et al. (2022) the semi-volatile fraction in exhaust emissions of diesel engines can be a substantial fraction (~35–90 %). While the semi-volatile part is formed by ions and organics the

solid part is mostly elemental carbon, so the use of DPF does not necessarily result in a strong decrease of total PNC (Giechaskiel et al., 2022).

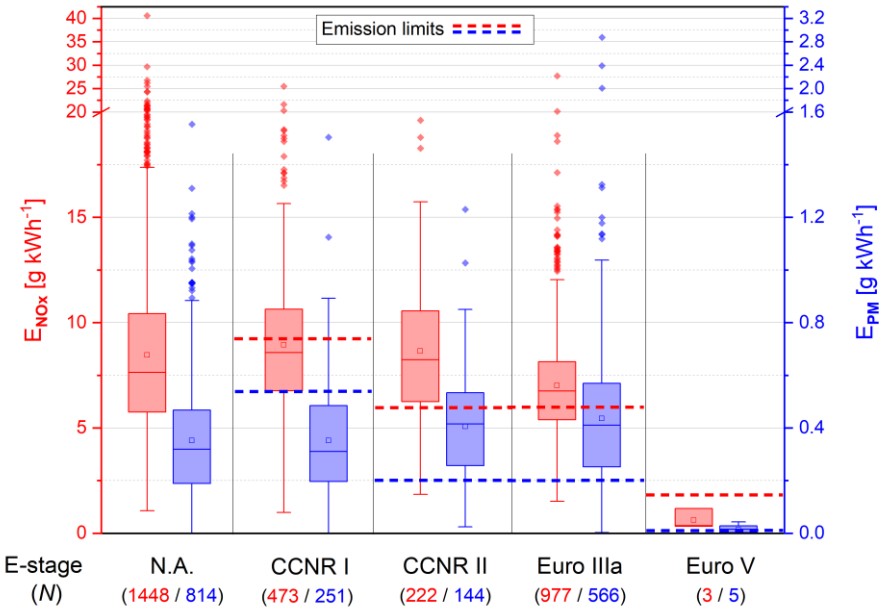

**Figure 5:** Comparison of NO$_x$ and PM emission factors ($E$) from this study with existing limits (red dashed line for NO$_x$, blue for PM) per emission stage category (see Table S1): CCNR I standard (valid from 2003 on), CCNR II (2007), Euro IIIa (2007) and Euro V (2020).

Boxplot: box = 25 to 75 % quantiles, whisker = inter-quartile-range (IQR), square = mean, solid line = median. The numbers in parenthesis indicate the amount of NO$_x$ / PM peaks the box plot is based on. N.A. means information is not available from the ship database.

With the help of technical ship information data (see Sect. 2.2.3) we divided the observed ships into emission categories based on the limit the ship engines have to comply with (see Fig. 5 and Table S1). For vessels fulfilling CCNR I standard we derived

a value of (8.9±3.2) g (kWh)$^{-1}$ for NO$_x$ which agrees very well with the corresponding emission limit of 9.2 g (kWh)$^{-1}$. Analysis of CCNR II vessels resulted in a mean emission factor of (8.7±3.2) g (kWh)$^{-1}$, which is slightly above the limit of 6 g (kWh)$^{-1}$.



For Euro IIIa vessels with a mean emission factor of with $(7.0\pm2.5)$ g $(kWh)^{-1}$ a better agreement is achieved. For Euro V vessels the measured emission factor of $(0.6\pm0.5)$ g $(kWh)^{-1}$ is a factor of 3 below the limit, although based on very few data points, as we captured only one ship equipped with such a modern engine. With regard to PM emission factors, CCNR I vessels are well below the limit (Fig. 5) whereas for CCNR II and Euro IIIa vessels the corresponding limit is exceeded by a factor of 2 on average. This could indicate that real-world particle emissions might not as efficiently be reduced as under test bench conditions. However, we have to keep in mind the relatively large uncertainty arising from the estimate for the particle density (Sect. 2.2.2) and the substantial fraction of measured semi-volatile particles discussed above. The number of measurements for Euro V vessels was very limited but results suggest that requirements were fulfilled through the application of DPF.

We also detected a small number of ships (~ 5 %) where the plume was characterized by very low BC levels (often below the detection limit), sometimes in combination with an extraordinary high $NO_2$-to-$NO_x$ ratio (up to 50 %). This indicates the use of aftertreatment systems with catalyst and DPF, where high $NO_2$ levels in the exhaust are favored for the filter regeneration. Unfortunately, we do not have any information about potential filtering technology used on board (only about the engine itself).

### 3.4 Dependence of $NO_x$ emissions on ship parameters

The large variability in $NO_x$ emission factors ($E_{NO_x}$) observed in this study is consistent with other field studies dealing with inland shipping (Kurtenbach et al., 2016; Krause et al., 2022) and raises the question of potential influencing factors. In the following the effect of operating conditions (direction of travel, speed through water) and ship parameters (ship type, engine characteristics) on emissions are brought into context. Generally spoken, the dependence of emissions on ship parameters and atmospheric conditions is very complex and variable from ship to ship. For inland ships there is little information available in the literature whereas for sea-going ships detailed studies or reviews are e.g. provided by Celik et al. (2020) and Grigoriadis et al. (2021).

At the Rhine in Worms we observed a decrease in $E_{NO_x}$ at higher engine loads, i.e. for upstream ships compared to downstream ships (Fig. 6a). In contrast, the $NO_2$-to-$NO_x$ ratio was very similar, indicating no significant differences in plume age. With regard to the vessel speed (Fig. 6b) we noticed a slightly decreasing $E_{NO_x}$ with increasing STW (except for very low speeds). A high steady state engine load is commonly related to a higher combustion temperature and a more complete combustion process, resulting in in a positive correlation between STW/engine load and $E_{NO_x}$, especially for sea-going ships with high-powered motors (Cappa et al., 2014; Celik et al., 2020). On the other hand, a higher engine load can lead to a reduction in the air-to-fuel ratio and thus in the oxygen content, resulting in reduced $NO_x$ formation, which is assumed to have a major effect for inland ship engines in this study. Simultaneously, the $NO_2$-to-$NO_x$ ratio tends to decrease with increasing STW (Fig. 6b). This can be traced back to the higher combustion temperatures leading to enhanced NO formation and is in line with observations for sea-going ships (Celik et al., 2020). For inland ships, the generally weak dependence of emission factors on engine load under normal operating condition observed in this study is in line with observations of (Kurtenbach et al., 2016; CLINSH, 2022) and suggests the existence of other influencing parameters.



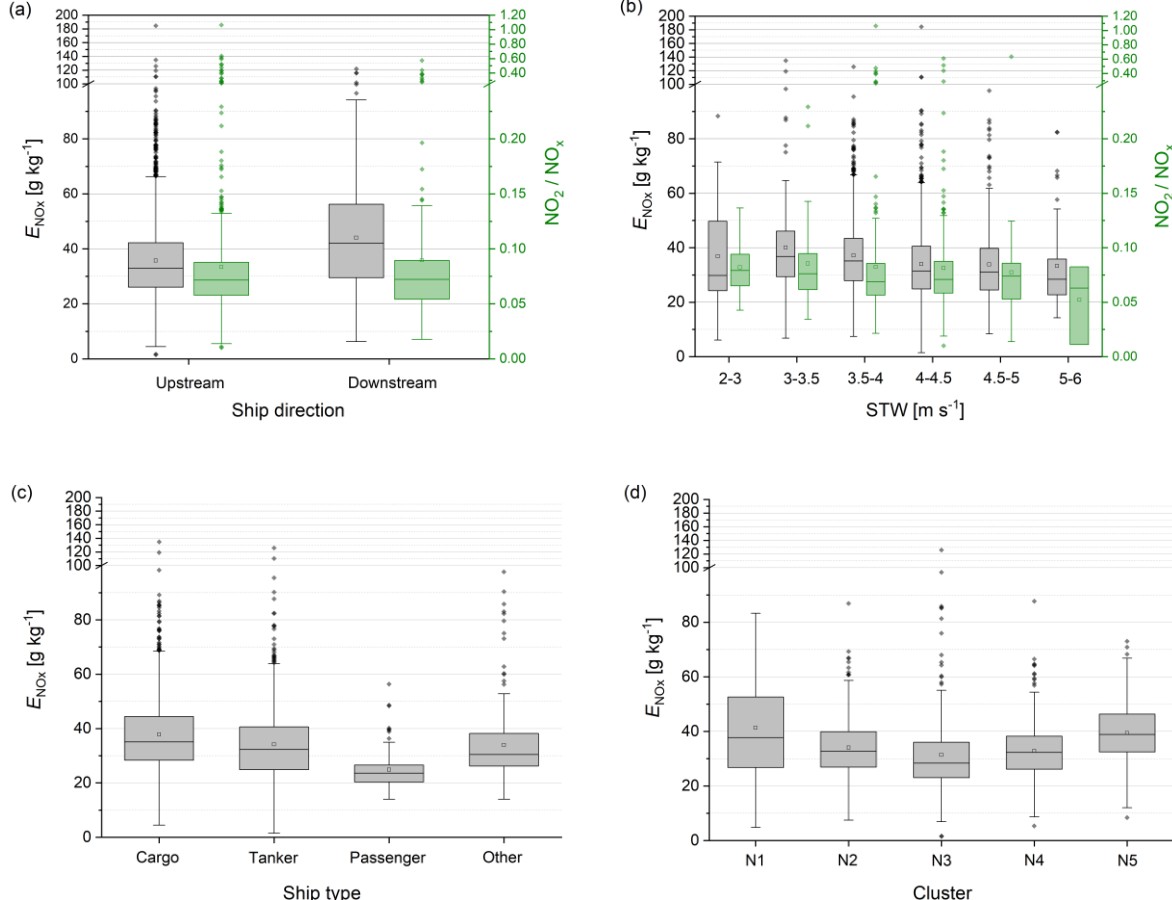

**Figure 6:** Influence of ship parameters on NO$_x$ emission factor ($E_{NOx}$) and NO2-to-NO$_x$ ratio: (a) travelling direction, (b) speed through water (STW), (c) ship type and (d) clusters based on engine characteristics (see Table 3). For better readability the scaling of the y-axis was adapted for $E_{NOx}$ > 100 g kg$^{-1}$ respectively NO2/NO$_x$ > 0.25.

**Table 3:** Overview of clusters based on engine characteristics and ship dimensions together with average NO$_x$ emission factors ($E_{NOx}$).

| Cluster | Motor age[a] | Power[b] [kW] | RPM[c] | Ship length[d] | Ship width[e] | $E_{NOx}$ [g kg$^{-1}$] | NO2/NO$_x$ [%] | $N_{peaks}$ |
|---------|-----------|--------------|--------|-------------|------------|----------------------|--------------|-----------|
| N1 | old | low | low | low | low | 41.4±18.2 | 0.06±0.03 | 92 |
| N2 | medium | medium | high | high | high | 34.1±11.5 | 0.10±0.10 | 244 |
| N3 | medium | low | high | medium | medium | 31.4±13.4 | 0.08±0.05 | 350 |
| N4 | medium | high | high | medium | medium | 32.8±9.8 | 0.08±0.03 | 450 |
| N5 | medium | high | medium | medium | medium | 39.4±11.9 | 0.07±0.04 | 145 |

[a]Motor age (mean): old (1970-1972), medium (2011 - 2013)
[b]Power (mean): low (670-720), medium (1090-1110), high (1170-1200)
[c]Revolutions per minute (RPM) of motor (mean): low (550-570), medium (940-970), high (1600-1780)
[d]Ship length (mean): small (90-95), medium (105-125), large (170-175)
[e]Ship width (mean): small (9-10), medium (11-12), large (14)



In addition to dynamic operating conditions, inland vessels were distinguished by their type, whereby cargo, tanker and passenger ships are of major significance on the Upper Rhine. Figure 6c shows, with an average of 33 g kg$^{-1}$, cargo ships had the highest mean NO$_x$ emission factor, while it was 10 % lower for tankers and 35 % lower for passenger ships. A possible explanation is a more frequent modernization for the latter type of ships. This finding is consistent with Pirjola et al. (2014)

and Kattner (2019) and a similar tendency was observed for sea-going ships in the Baltic and North Sea (Beecken et al., 2014). Excluding passenger ships, the influence of ship type and operating conditions on the $E_{NOx}$ can be considered rather small and is not sufficient to explain the observed variability in our dataset.

As a consequence, we investigated the effect of engine characteristics like motor power or revolutions per minute (RPM) by performing a cluster analysis (see Sect. 2.3.4), with the five clusters based on 1281 ship peaks described in Table 3. The effect

on NO$_x$ emission factors is illustrated in Fig. 6d and reveals that for cluster N1, containing ships equipped with old motors (mean year of construction 1972), low power (< 700 kW) and low RPM (< 600), NO$_x$ emissions factors were enhanced by 20 to 30 % compared to clusters N2, N3 and N4. Increased NO$_x$ emission factors were also observed for medium aged motors with high power but medium RPM (cluster N5). This is consistent with the positive correlation between power and NO$_x$ emissions found for sea-going ships (Diesch et al., 2013). In contrast, low emissions were observed for relatively new motors

with high RPM (clusters N2–N4), closest to the motor type used in modern trucks. Our findings are consistent with literature studies for sea-going ships where motors with high power and low RPM are associated with larger emissions compared to motors with low power and high RPM (Grigoriadis et al., 2021). This is mainly due to a longer residence time in the combustion chamber at low RPM, leading to an enhanced NO$_x$ content in the exhaust gas (Reif, 2012).

### 3.5 Dependence of particle emissions on ship parameters

The mean particle size distribution (PSD) from 6 to 520 nm based on 1822 ship peaks showed a unimodal size distribution for up- and downstream vessels with a geometric mean particle diameter of 52±23 nm and a mode diameter ($D_{mode}$) of 66±35 nm. On average, the majority of particles was found in the Aitken mode which is consistent with results from Pohl et al. (2017) for inland vessels on the Lower Rhine. Figure 7a describes the relative abundance of the mode diameter ($D_{mode}$), i.e. the channel where the highest particle count was measured. This distribution has a bimodal appearance, although each ship's individual

peak was normally characterized by a unimodal size distribution with the mode diameter being highly variable from ship to ship (Fig.7b, A–C). For both travelling directions there are ships which emit the majority of particles in the range of 10–40 nm (nucleation mode) and ships where this maximum lies in the range of 60–110 nm (Aitken mode). For a small number of ship plumes also both modes existed at the same time with either the nucleation (Fig. 7b, D) or the Aitken mode (Fig. 7b, E) being dominant. For diesel engines bimodal distributions are typical, but it depends on various parameters which mode exists or

dominates (Kittelson, 1998). Particles in the nucleation mode are mostly hydrocarbons from lubrication oil, unburnt fuel, water vapor or sulfur particles. Old inland ship engines and engines with low RPM (two-stroke diesel) typically show a pronounced nucleation peak. The accumulation mode consists mainly of ash (elemental carbon, EC) and is a typical feature of car diesel



engines. In this regard it is important to mention that the actual size distribution can be highly individual depending on the engine and the DPF used.

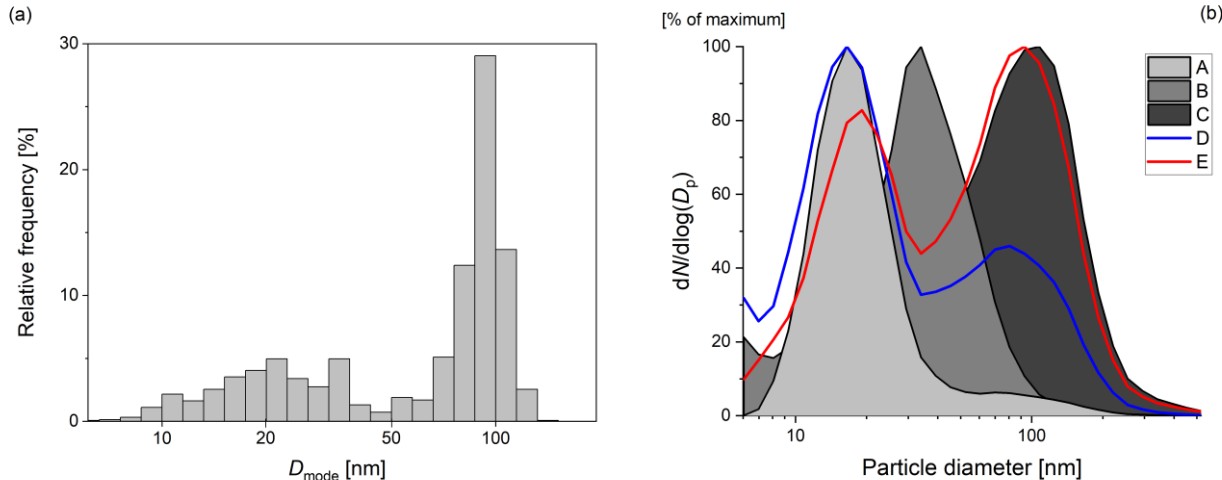

**Figure 7:** (a) Relative abundance of the particle mode diameter ($D_{mode}$) for upstream ships based on 1503 peaks. (b) Examplary particle
size distribution (PSD) from five ships (labeled A–E).

A comparison of inland ships with ocean-going vessels is only reasonable to a limited extent, as the larger sulfur content affects the total number of particles and results in a pronounced nucleation mode (Celik et al., 2020; Diesch et al., 2013; Pirjola et al., 2014; Zhou et al., 2019). This is due to the conversion of $SO_2$ to $SO_3$ in the catalyst and reaction with water molecules forming gaseous sulfuric acid (Karjalainen et al., 2014). The $D_{mode}$ of ships with low sulfur content is usually between 30 and 60 nm which is similar to our study. With higher sulfur content, the particles become slightly larger (Zhou et al., 2019). Significant increases in PNC at a diameter > 250 nm were not measured in other studies (Diesch et al., 2013; Beecken et al., 2015; Ausmeel et al., 2019; Kuittinen et al., 2021), i.e. particles are exclusively detected in the UFP range, consistent with emissions from inland vessels in this study. Depending on ship characteristics both uni- and biomodal size distributions were observed by others (Jonsson et al., 2011; Diesch et al., 2013; Pirjola et al., 2014; Beecken et al., 2015). Furthermore, for when comparing inland and seagoing vessels, a higher relative speed or a higher engine load leads to higher emissions for inland vessels, while the opposite is the case for seagoing diesel engines. Here, cleaner combustion takes place when the ships are operated under high load, resulting in lower emissions (Pohl et al., 2017).

A more reasonable comparison is possible with diesel-powered road transport, due to the same fuel. Several studies show a predominance of UFP (Chatain et al., 2021; Kwasny et al., 2010) , similar to emissions from inland shipping in this study. The main difference is the use of catalysts and particulate filters in road transport which are rarely applied in inland ships so far. This alters the PSD and PNC in the exhaust gas and results in a varying manifestation of nucleation and accumulation mode. Older cars without particulate filter, partly with sulfur-containing diesel cause a bimodal peak (Giechaskiel et al. (2005). For





newer vehicles with particulate filters, a reduction of particulates in the nucleation mode was observed, resulting in a unimodal PSD (Braisher et al., 2010; Li et al., 2018).

In this study PNC, PM and also PSD were influenced by different ship-related parameters. In the following, the impact of STW, ship type, engine power, revolutions per minute (RPM) and year of construction was analyzed. Figure 8a shows that the

PNC emission factor ($E_{PNC}$) for downstream ships is higher than for upstream ships. At the same time particles of downstream ships have a slightly smaller mode diameter ($D_{mode}$) and geometric mean diameter (GMD) than for upstream ships. This may result from differences in the fuel consumption. Ships tend to use more fuel when travelling upstream, which means increased particle generation, more particles per volume in the exhaust pipe and thus in the plume. In a high-density plume, nucleation mode particles can coagulate faster on the way from the exhaust to the point of measurement, resulting in an increased mean

particle diameter. With a BC fraction of 38 % for upstream ships and 16 % for downstream ships our results indicate that a higher engine load leads to an increase in BC emission. This is in contrast to sea ships where measurements show a decrease in BC with an increase in combustion efficiency (Celik et al., 2020). From multi-wavelength analysis of the AE33 we derived a mean fractional contribution from biomass burning of ~10 %, which might be indicative for organic aerosol components formed in biofuel combustion (Sandradewi et al., 2008).

No clear correlation between STW and $D_{mode}$ was found (Fig. 8b). The highest $E_{PNC}$ was observed for ships with a speed of 2–3 m s$^{-1}$ and 5–6 m s$^{-1}$, i.e. exceptionally low or high engine loads tend to result in higher emission factors. These groups also contained the lowest mean $D_{mode}$ with 47 nm for 2–3 m s$^{-1}$ and 56 nm for 5–6 m s$^{-1}$. The classification into ship types showed the highest mean $E_{PNC}$ for passenger ships (and thus the opposite trend than for NO$_x$, see above) whereas $D_{mode}$ (and $E_{BC}$, see SI) was the lowest (Fig. 8c).

For a classification based on ship size, ships were often divided into ship classes in past studies (Kurtenbach et al., 2016; Krause et al., 2022). This division showed that smaller ships were tendentially characterized by larger $E_{PNC}$ and lower $D_{mode}$ (see Table S5). But small ships are usually older and have lower engine power and RPM. Thus, it is probably not the length and width of the ships included in the ship classes that is decisive for $D_{mode}$ and $E_{PNC}$, but the engine. Based on these findings, a cluster analysis using particle data from upstream vessels (683 peaks from 341 ships) and corresponding engine data was

done (see methods). Figure 9d shows the frequency distribution of $D_{mode}$ and $E_{PNC}$ for each cluster. Cluster P2 was characterized by old engines, with a mean year of construction of 1970, low RPM, low power, and small width/length (Table 5). It shows the smallest particles with a mean $D_{mode}$ of 46 nm. Also, $E_{PNC}$ was highest for this cluster with $7.8 \times 10^{15}$ particles kg$^{-1}$. This means that data from engines of a high age with low power and low RPM were characterized by high $E_{PNC}$ and comparably small particles. For cluster P5, which reflects ships with modern engines, $E_{PNC} = 2.1 \times 10^{15}$ g kg$^{-1}$ is low compared to the other

clusters and the $D_{mode}$ mean of 82 nm is also highest. This could be due to a more complete combustion by the newer engine and thus a less developed nucleation mode. Thus, it can be concluded that the different PSD and PNC result from different engine characteristics, in particular the year of construction, power and RPM.



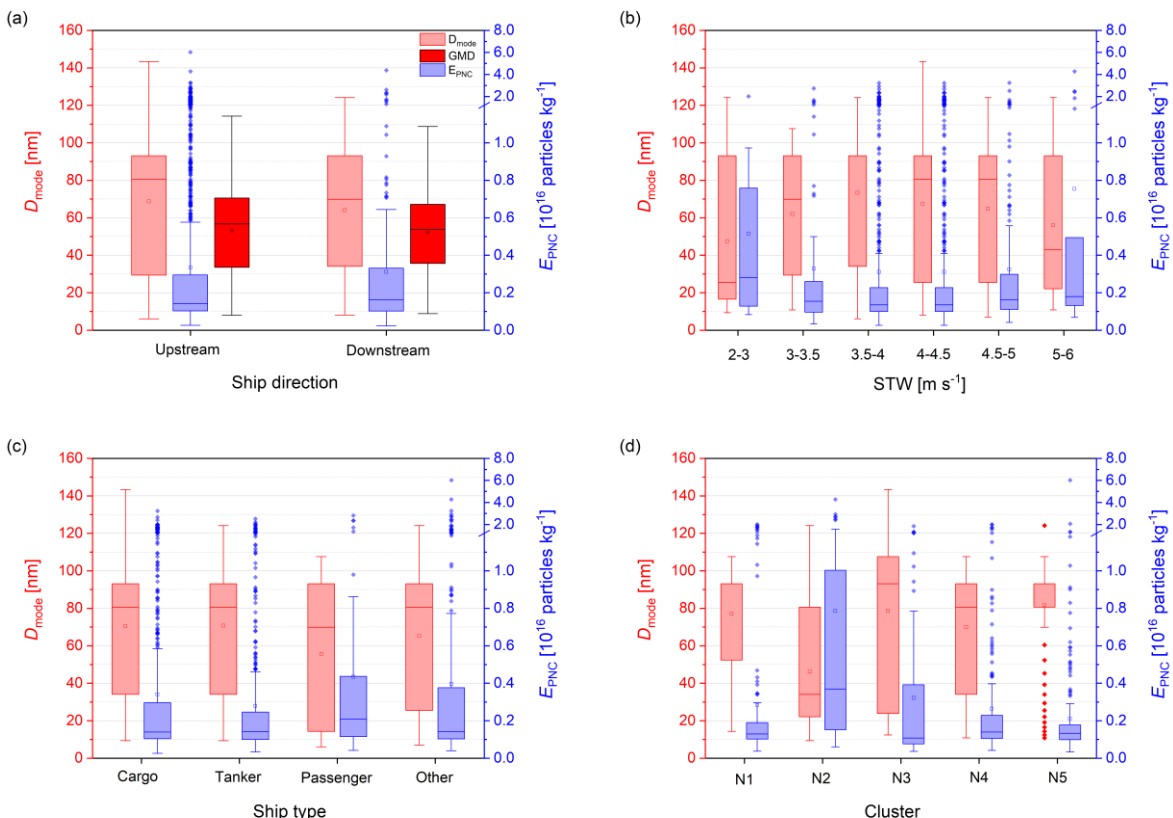

**Figure 8:** Influence of ship parameters on PNC emission factor ($E_{PNC}$) and particle mode diameter ($D_{mode}$): (a) travelling direction, (b) speed through water (STW), (c) ship type and (d) clusters based on engine characteristics (see Table 4). For better readability the scaling of the y-axis was adapted for $E_{PNC} > 1.2 \times 10^{16}$ particles kg⁻¹.

**Table 4:** Overview of clusters based on engine characteristics and ship dimensions together with average PNC emission factors ($E_{PNC}$).

| Cluster | Motor age[a] | Power[b] (kW) | RPM[c] | Ship length[d] | Ship width[e] | $E_{PNC}$ (particles kg⁻¹) | $D_{mode}$ (nm) | $N_{peaks}$ |
|---------|--------------|---------------|--------|----------------|---------------|-----------------------------|------------------|-------------|
| P1 | medium | medium | high | high | high | $(2.8\pm4.3) \times 10^{15}$ | 77±32 | 126 |
| P2 | old | low | low | low | low | $(7.8\pm9.5) \times 10^{15}$ | 46±30 | 46 |
| P3 | medium | high | medium | medium | medium | $(3.2\pm4.3) \times 10^{15}$ | 79±42 | 60 |
| P4 | medium | low | high | medium | medium | $(2.6\pm3.4) \times 10^{15}$ | 70±31 | 177 |
| P5 | medium | high | high | medium | medium | $(2.1\pm4.1) \times 10^{15}$ | 82±28 | 274 |

[a]Motor age (mean): old (1970-1972), medium (2011 - 2013)
[b]Power (mean): low (670-720), medium (1090-1110), high (1170-1200)
[c]Revolutions per minute (RPM) of motor (mean): low (550-570), medium (940-970), high (1600-1780)
[d]Ship length (mean): small (90-95), medium (105-125), large (170-175)
[e]Ship width (mean): small (9-10), medium (11-12), large (14)

Unfortunately, there are no other studies explicitly addressing the impact of engine characteristics of inland ships on particle emissions. Studies of inland ship emissions related to PM show a positive correlation with power or speed (Busch et al., 2020).



However, analyzed vessels at berth show a negative correlation with power, but a positive with age (Busch et al., 2021). In our study, no influence of these parameters on PM was evident. For sea-going ships an increase in PNC for diameters > 100 nm is associated with decreasing speed and attributed to incomplete combustion (Diesch et al., 2013). Other factors influencing the PSD were, similar to our results, the year of construction and the power (Diesch et al., 2013; Pirjola et al., 2014). The differences between the individual clusters for both particles and $NO_x$ (see Sect. 3.4.) suggest that motor age, RPM and engine power have the largest effect on the emission factor of inland ships, while ship size and STW only play a minor role under normal operating conditions.

## 4 Conclusions

We investigated the contribution of inland shipping to air pollution based on immission measurements at the river Rhine in Worms, Germany. Measurements were carried out over the course of more than one year at two measurement stations in direct vicinity to the Rhine (one directly above the lane, one on the river banks) and represent the first long-term on-shore measurements available for the Upper Rhine. By combining a peak analysis algorithm with AIS data, the monthly average contribution from inland shipping ($NO_x$, $NO_2$, PNC, UFP, $PM_1$ and BC) was derived for the two measurement stations and put into relation to atmospheric background levels. For $NO_2$ a monthly mean contribution of 7 % in addition to the local background concentration was calculated for BRI and 3 % for RIV. Directly above the shipping lane at BRI shipping led to an increase of 50 % in $NO_x$, whereas PNC was increased by 10 %, $PM_1$ by 4 % and BC by 15 %. The majority of particles was measured in the UFP range (~75 %), with a dominance of the Aitken mode. The contribution from shipping was highly variable from day to day, depending on prevailing wind conditions. Although not being representative for other parts of the city of Worms, our measurements provide a useful estimate on the typical exposure of people living or working very close to the Rhine.

The applied peak analysis mechanism can easily be transferred to other locations as long as $CO_2$ and AIS data are collected together with the compound of interest (e.g. $NO_x$ or PM). Prerequisites facilitating the analysis are that the station is close to the river (peaks distinguishable from noise), the traffic density is not too high (no peak overlap) and the variability in the background signal (arising from non-ship emission sources like road traffic) is low in time scales of minutes. The advantage of our method is that is does not rely on dispersion modelling and that energy-related emission factors are transferable to other locations, fleets and operating conditions (although the fuel consumption has to be carefully estimated). This enables the application in emission inventories and modelling studies to more accurately calculate the exposition of people to air pollution along inland waterways.

The observed PSD of inland ships and the large proportion of UFP highlight the necessity to integrate UFP measurements in future studies, especially with regard to the recommendations for action of the WHO, where a more in-depth research of UFP is requested (WHO, 2021). In particular optical measuring devices that are not able to detect particles < 100 nm are not suitable to be applied in the specific case of shipping emissions and may bias the results when operated alone. This study also underlines



the necessity of considering inland vessels separately, since results from studies about sea-going ships or road transport cannot easily be transferred due to differences in motor dimensions, driving conditions, fuel sulfur content and particle filtering technology, although the dominance of UFP is a common feature.

From single ship plumes we derived realistic emission factors for the fleet on the Upper Rhine in Worms under real-driving conditions, complementing existing studies at the Lower Rhine. Reported $NO_x$ emission factors were on average lower than at the Lower Rhine and approximately in line with requirements set by the CCNR and the EU, whereas PM emission factors were slightly increased. For the small number of ships with modern Euro V diesel engines and aftertreatment system passing the station, we derived very low $NO_x$ and PM emission factors underlining their emission reduction potential. A detailed analysis with respect to ship direction, STW, ship type and ship class revealed that the large observed variability in emission factors could hardly be accessed by classification schemes commonly used in the literature. Using additional technical data about the ship engines, we figured out that both gaseous and particulate emissions were strongly dependent on engine characteristics (especially RPM and power). Particularly, old ships with low RPM and power caused high emission factors for both $NO_x$ and PNC, whereby the PSD was characterized by a small particle mode diameter. These findings underline the need to overcome limitations in data accessibility, as the use of emission factors related to engine-specific data (which are not automatically sent via AIS) can help to improve impact calculation of inland shipping in modeling studies. In addition, the results illustrate the potential of long-term on-shore measurements to monitor future effects of fleet modernisation, i.e. emission reduction by replacing or retrofitting engines with DPF or exhaust gas aftertreatment systems.

For future studies we aim on the additional measurement of particle composition (e.g. via AMS) to derive a realistic estimate for the particle density. The existing data basis could also be extended by further measurements at different Rhine sections and along other relevant waterways in Europe in order to capture various fleet compositions and driving conditions. For a more detailed investigation of the plume dispersion in cities along rivers a monitoring network with sensors placed at different distances to the shipping lane in combination with measurements on-board and a small-scale modelling approach would be of significant value.

**Data availability**

Datasets from this study are archived at https://doi.org/10.5281/zenodo.7896435 (Eger et al., 2023).

**Author contributions**

TM and PE wrote the manuscript with contributions from AZ and LD. Measurements in Worms were conducted by TM and PE with AZ supporting the setup of the measurement equipment. TM and PE performed the data analysis with TM focussing on particles and the influence of ship and engine parameters on emissions, and PE focussing on $NO_x$, the peak detection algorithm and the derivation of emission factors.



**Competing interests**

The authors declare that they have no conflict of interest.

**Acknowledgements**

We thank Svenja Sommer (Bundesanstalt für Gewässerkunde) for her assistance in the recording and evaluation of AIS data.
We would also like to thank the Landesamt für Umwelt Rheinland-Pfalz (in particular Michael Weißenmayer, Margit von
Döhren and Eduard Neufeld) for the meteorological data from the measurement station in the city of Worms and auxiliary
$NO_2$ and BC measurements in the initial phase of the measurement campaign. We are grateful to the Landesbetrieb Mobilität
Rheinland-Pfalz (in particular Hubertus Darmstadt and Gerald Krallinger) for providing the rooms inside the Nibelungen
bridge as a measurement spot. We also thank the Wassersportverein Worms (in particular Rudolf Schöpwinkel) for provision
of the club area to install our measurement container and weather station at the river banks. We are thankful to Airyx (in
particular Stefan Schmitt, Denis Pöhler und Richard Brenner) for their contribution to the setup of the ICAD measurement and
the inlet construction at the bridge. We would also like to thank the Wasserschutzpolizei Gernsheim (in particular Michael
Spahn and Andreas Dickes) for providing a measurement spot for preliminary measurements at the Rhine in Gernsheim.

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
