# Peer review of "Table S1: Emission limits set by the CCNR (CCNR, 2022) and the EU (European Union, 2016) in comparison with mean emission factors from this study (measured at BRI)."

_EGUsphere, 2023_

## Referee Comment (RC2)

Review on article:

Measurement of inland ship emissions and their contribution to NOx and ultrafine particle concentrations at the Rhine

Philipp Eger et al.

The authors present the results of a comprehensive study on the differentiated assessment of inland shipping emissions on the Upper Rhine near Worms in Germany. They use a wide range of measurement techniques to detect gaseous ($CO_2$, NOx and $O_3$) and particulate (PNSD, PNC, PMx, soot) air pollutants. Two sites have been selected for the measurements, allowing different scenarios to be mapped. One site was located on a bridge in order to record the plumes from passing ships close to the source. The second site was chosen directly on the banks of the Rhine. In this way, it is possible to determine the level of emissions that could affect people living near the Rhine. Particularly noteworthy is the methodology developed to identify individual ship plumes. The algorithm used avoids overlapping plumes, which can be caused by several ships passing at the same time. As a result, only clearly identifiable ship plumes are included in the evaluation. This results in a significantly reduced number of evaluable ship plumes and also reduces the number of individual ships in the composition of the shipping fleet. At the same time, the quality of the subsequent allocation and classification is significantly improved. In particular, the continuous long-term measurements over a period of one year provide a good picture of the emissions of the shipping fleet in this part of the Rhine. In addition, the emission factors can be calculated under real conditions, leading to a better understanding of the impact on inland navigation. This work represents a solid contribution and, in part, a new scientific approach to the measurement and characterisation of emissions from inland navigation under real conditions. The work is recommended for publication by the peer reviewer. The following suggestions may be incorporated into the authors' opinion.

P3 L21:

*…high temporal resolution of ~1 s…*

Maybe one can mention, that the SMPS has a different and longer temporal resolution for a whole scan of the size range. Additionally, one could also explain the "problem" with scanning devices as a SMPS with a moderate sampling time. The assumption with a scanning device as the SMPS is that the aerosol spectrum does not change much over the time of a scan. However, this can occur with passing ships and short-term increases and thus lead to a distorted PNSD.

P4 L11

*…Instrument-specific sampling lines of 4-5 m length…*

It seems that the calculated particle loss under 10 percent is relatively low. I would expect a higher particle penetration at this length of the sampling line. Did you use separeted sampling lines or did you use one sampling line with a higher volume flow and a manifold leading to the individual measuring devices?

P4, L12

*…to enable an undisturbed incoming flow.*

Doesn't the bridge itself generate turbulence that can contribute to influencing the wind field at the measurement site? Are downwind eddies possible that carry road traffic emissions down to the measurement site and superimpose the ship plumes as well?

P5 L5

*…to avoid strong interferences from road traffic.*

You have chosen the locations to also avoid the influence of traffic related air pollutants. I am not familiar with the local conditions, but a look at the Nibelungen Bridge shows that this is a double bridge with two lanes each. What traffic volume can be expected there? Is there rush hour and congestion with traffic jams on the bridge? Especially with winds from northern directions, lee vortices could transport the TRAPs to the sampling point.

P6 Table 1

Here the temporal resolution from the AIS signals is 1 s. To the best of my knowledge, an inland vessel sends a data set only every 10 s, depending on the current movement status.

P12 L20-21

*…further results […] refer to this instrument.*

This sentence is somewhat confusing, since in the coming chapters the results on RIV site will also be reported, which, however, were measured with the SMPS.

P12 L26-27

The study by Pohl et al. was performed in Duesseldorf. So please change Upper to the Lower Rhine.

P14 L16

*…as well as modern ships with exhaust after treatment…*

With regard to the CLINSH project. Weren't up to 40 ships retrofitted with downstream exhaust aftertreatment systems? Are the data or names of the ships available the authors to specifically read them out in their data set in order to be able to better scale up the positive effect of the emission reduction? This would be a good contribution, especially in view of the continuing increase in shipping traffic in the future.

P21 L10

*With a BC fraction of 38 % for...*

It is (for me) not clear to which correlation the value is. Can you please more specify this. Is it $BC_{880\,nm}$ to total BC?

P21 L14

The proportion coming from biomass burning is mentioned here as about 10 % from biofuel combustion. Could it be a possible reason that the analyzed probe isn't just from ships because you also measure the background were also particles coming from wood fires, cigarette smoke, etc. Maybe there could be a hint, if the amount of bb is higher during the wintertime due to fireplaces?

P21 L25

*…(see methods).*

Please refer to the chapter.

---

## Author Comment (AC1)

**Reply to RC1**

We thank the referee for the review of our manuscript and the valuable comments and suggestions. In the following the referee's comments are repeated (in black) along with our replies (in blue) and changes made to the text in the revised manuscript (in red). Page and line numbers refer to those in the preprint version.

**General comments:**

This manuscript describes the results of real-world emissions measurements made for ~4700 inland ship passages in the Upper Rhine valley for over a year. This work presents $NO_x$ and PM fuel-based emissions factors measured above the ships and on the riverbank and relates them to AIS-obtained ship data. The authors found that the shipping emissions increased $NO_x$ and PM for those localized areas from 3-50%. Additionally, their calculated emission factors showed correlation to engine characteristic such as RPM and engine age/category. This work showed the great importance of real-world measurements to validate existing emissions standards and inventory and demonstrate correctly functioning or malfunctioning emission control technologies. This reviewer agrees with the authors that this long-term measurement work is critical to creating reliable estimates of shipping emissions contributions and could be applied to multiple emission source sectors.

However, the authors do not consider other related work done outside of Europe on understanding the emissions and engine load relationship (see recommendations below).

We agree with the referee that related work outside Europe should be considered. We included the suggested publications in the corresponding sections (see specific comments below).

We also added the regional study of Buffaloe et al. (2014) to the paragraph about sea-going ships in the introduction.

The impact of sea-going ships (and non-European inland ships) on air quality in coastal and port areas was more frequently investigated (Moldanová et al., 2009; Jonsson et al., 2011; Diesch et al., 2013; Alföldy et al., 2013; Beecken et al., 2014; Buffaloe et al., 2014; Pirjola et al., 2014; Beecken et al., 2015; Seyler et al., 2017; Cao et al., 2018; Celik et al., 2020; Jiang et al., 2021; Kuittinen et al., 2021; Schwarzkopf et al., 2022), but due to different fuel types […]

Additionally, there could be additional forecasting or recommendations provided by the author on how to transition the older shipping fleet to newer engines and what the emission and exposure benefits may be from this transition.

Following this idea, we compared different future emission scenarios (e.g. assuming a complete transition to Euro V emission standard) to quantify the impact of the engine emission category on the mean emission factor and the additional contribution from shipping (see specific comments below).

Overall, this work provides well-described and illustrated novel scientific contribution and is recommended for publication with revision. Please review the specific line comments and recommendations below.

**Specific comments:**

*P10 L19-21 In the latter case, which was true for ~ 5 % of all ship plumes, the BC peak height was manually set to zero, marginally reducing the calculated mean emission factor.*

Integrating near-zero values is still recommended rather than manually setting to 0 in order to detect potential failing in the filter technologies on-board (especially if tracked over time).

We performed the analysis again, manually integrating all BC peaks that have not been automatically recognized by the peak finding algorithm. If there was a clear peak in $CO_2$ but no (or only a very small) increase above background level in the BC signal, the BC peak area resulted in a value close to zero. The text was modified accordingly.

In addition, missing peaks in BC were checked in detail to decide whether the peak was just not automatically recognized or if there was really no BC or only a very small amount present (e.g., due to applied particle filtering technology). In the latter case, which was true for ~ 5 % of all ship plumes, the BC peak area was integrated manually (resulting in a value close to zero). This way all the non-BC or very weak-BC emitters were included in the statistics as well, which led to a marginal reduction of the calculated mean emission factor resulting from the automatic peak finding routine.

*P12 L9-12 This value is also typical for diesel engines without exhaust gas aftertreatment systems, assumed to constitute the largest portion of the inland fleet (see Sect. 2.2.3), whereas the use of catalysts and diesel particulate filters (DPF) usually results in a somewhat higher ratio of 0.25– 0.30 (Kurtenbach et al., 2016).*

Additional sources exist detailing the use of onboard emission control technologies that may be cited here (see suggestions below)

We added the suggested literature (Nuszkowski et al., 2009; Gysel et al., 2015; Sugrue et al., 2022) and gave more details on the effect of onboard emission control technologies like selective catalytic reduction (SCR) and diesel particulate filter (DPF).

The use of catalysts and diesel particulate filters (DPF) usually results in a somewhat higher ratio of 0.25–0.30 (Kurtenbach et al., 2016), whereas selective catalytic reduction (SCR) technology decreases it to 0.01. Applied to marine engines SCR and DPF were found to efficiently reduce $NO_x$ and PM emissions by up to 90 % (Nuszkowski et al., 2009; Gysel et al., 2015; Sugrue et al., 2022).

*P13 L12-14 Monthly average concentrations of $NO_x$ were increased by 6.9–14.3 μg $m^{-3}$ at BRI (above the shipping lane) and by 1.0–2.2 μg $m^{-3}$ at RIV (river bank). This corresponds to an additional burden of 24–68 % respectively 6–11 %, relative to extrapolated background levels without the shipping contribution.*

Though noted later in the manuscript, it is important to clarify that these increases are extremely localized and dependent on meteorology. Additionally, comments regarding the number of people living along the river area could be noted to increase the significance of this finding.

We agree that emphasizing the proximity of the measurements to the shipping lane is important and added the following text:

We note that the additional contributions reported here were observed in close proximity to the shipping lane and were strongly dependent on meteorological conditions.

We cannot give detailed numbers on how many people are living within 50–100 m distance to the shipping lane. But we think that this is a typical distance for the waterfront in cities along the Rhine at which people are commonly breathing air for a longer period of time (walking, eating, working, living).

In the case of RIV the derived values at a distance of 50–100 m to the shipping lane are in particular representable for the exposure of people staying here for an extended period of time. This includes walking, doing sports or having lunch, as well as company buildings and houses with gardens located close to the riverside. A comparable exposure can be expected on the river banks of other cities along the Rhine, albeit strongly depending on meteorological conditions and ship traffic density.

*P15 L7-8 This again indicates potential differences in fleet composition, load status and operating conditions between Lower and Upper Rhine.*

This point needed further clarification as the distinction between the lower and upper Rhine is not clear to the reviewer.

The main difference is that the ship traffic density on the Lower Rhine is generally higher than on the Upper Rhine, impacting operating conditions. As different goods are transported on different Rhine sections, there are also potential differences in the fleet composition and load status. These parameters can influence the measured $NO_x$ emission factor. The text was modified accordingly.

This indicates potential differences in parameters like fleet composition, load status and operating conditions as the ship traffic density on the Lower Rhine (Duisburg) is much higher than on the Upper Rhine (Worms) and as different goods are transported on different Rhine sections. These parameters can influence the mean emission factor, underlining the need for measurements at various locations.

*P15 L21-23 The BC emission factor derived in this study ($E_{BC} = 0.5\pm0.3$ g kg$^{-1}$) is in the upper range of values reported for sea-going ships (Petzold et al., 2008; Diesch et al., 2013; Celik et al., 2020).*

Additional literature specifically on BC (and $NO_x$) emissions would help to frame this metric in comparison to other ship types or locations (see suggestions below).

Additional literature was added for $NO_x$ emission factors.

However, considerably lower values can be observed for marine ships operated in emission control areas and vessels equipped with SCR technology (Sugrue et al., 2022)

Additional literature was added for BC emission factors and the large variability dependent on various parameters was pointed out: Buffaloe et al. (2014) reported $E_{BC}$ = 0.31±0.31 g kg$^{-1}$ for sea-going ships with low-sulfur fuel at the Californian coast. Jiang et al. (2018) reported $E_{BC}$ = 0.05 – 1.84 g kg$^{-1}$, depending on engine load and fuel used. Schlaerth et al. (2021) measured 0.56±0.86 g kg$^{-1}$ for passenger ships and 0.54±0.56 g kg$^{-1}$ for tugboats in Southern California. Sugrue et al. (2022) measured 0.34±0.03 g kg$^{-1}$ for passenger vessels in San Francisco Bay.

The BC emission factor derived in this study ($E_{BC}$ = 0.5±0.3 g kg$^{-1}$) is in the upper range of values reported for sea-going ships (Petzold et al., 2008; Diesch et al., 2013; Buffaloe et al., 2014; Jiang et al., 2018; Celik et al., 2020; Schlaerth et al., 2021; Sugrue et al., 2022), although there is a large variability in the literature depending on ship type, engine type, engine load, fuel and operating conditions.

*P17 L5-6 This could indicate that real-world particle emissions might not as efficiently be reduced as under test bench conditions.*

This is a very important point and could be emphasized more so in the conclusion of this manuscript. Additional literature has commented / proved on this idea (see suggestions below).

Additional literature was included that underlines the suggestion above: Gysel et al. (2015) and Sugrue et al. (2022) concluded a less efficient $NO_x$ and PM reduction under low load operating conditions.

Potential reasons for this are malfunctions and unfavorable operating conditions, e.g. a low engine load (Gysel et al., 2015; Sugrue et al., 2022).

On p.17, l.7-8, we now write:

The number of measurements for Euro V vessels was very limited but results suggest that requirements were fulfilled through the application of DPF. This is in line with a ~90 % PM reduction observed in the literature for DPF technology applied in marine engines (Gysel et al., 2016)

An Additional statement was added in the conclusion:

This might be due to a decreased emission reduction efficiency in real-word operation compared with test bench conditions.

Although reported emission factors were slightly above the requirements for some engine categories, we want to stress again that our statement (p.17, l.5-6) is afflicted with a substantial uncertainty (arising from particle density and semi-volatile versus solid fraction) so that a limit exceedance cannot directly be concluded from our measurements. We slightly adapted one sentence in the abstract to make things clearer.

Comparison with emission regulation limits set by the Central Commission for the Navigation of the Rhine (CCNR) and the European Union (EU) showed that - within the uncertainty of our calculation method - mean energy-dependent emission factors under real-driving conditions were slightly exceeding those under controlled laboratory conditions.

And expanded the statement on p.16, l. 8-9 to differentiate between the semi-volatile fraction in terms of PNC and its smaller effect on PM.

According to Petzold et al. (2008) and Giechaskiel et al. (2022) the semi-volatile fraction in exhaust emissions of diesel engines can be a substantial fraction in terms of PNC (~35–90 %) whereby the contribution to PM is limited due to the small particle size.

*P17 L18-20 For inland ships there is little information available in the literature whereas for sea-going ships detailed studies or reviews are e.g. provided by Celik et al. (2020) and Grigoriadis et al. (2021).*

Additional literature is available on this topic (see suggestions below)

Additional literature was added: Buffaloe et al. (2014) and Sugrue et al. (2022).

For inland ships there is little information available in the literature whereas for sea-going ships detailed studies or reviews are e.g. provided by Buffaloe et al. (2014), Celik et al. (2020), Grigoriadis et al. (2021), and Sugrue et al. (2022).

*P21 L10-12 With a BC fraction of 38 % for upstream ships and 16 % for downstream ships our results indicate that a higher engine load leads to an increase in BC emission. This is in contrast to sea ships where measurements show a decrease in BC with an increase in combustion efficiency (Celik et al., 2020).*

This finding contradicts other literature that has seen a decrease in BC emissions as a function of engine load (as stated in the manuscript) but does not include suggested literature below.

Additional literature was added: Buffaloe et al. (2014) observed a decrease in BC emission factor for cargo ships operating close to the Californian coast. Jiang et al. (2018) found an increase in BC emission factor when increasing the load from 25 to 75 %. Schlaerth et al. (2021) observed increasing BC with increasing load. Results from different studies show that the comparison of emission factors measured on sea-going ships is complicated. Ferries, containerships, tug boats, cruise ships, pilot vessels are all operating under different conditions for very different time spans. We express this in the last sentence.

This is in line with studies by Jiang et al. (2018) and Schlaerth et al. (2021) on smaller marine engines (like passenger ships and tugboats) but is in contrast to Buffaloe et al. (2014) and Celik et al. (2020) who measured a decrease in BC with increasing engine load. The large variability in BC emission factor reported in the literature reflects the influence of engine type, fuel type and operating conditions.

*P24 L7-8 For the small number of ships with modern Euro V diesel engines and aftertreatment system passing the station, we derived very low $NO_x$ and PM emission factors underlining their emission reduction potential.*

Expanding on this idea and modeling out how the next generation fleet will improve emission and exposure levels would greatly contribute to this manuscript

Following this idea, we compared different future emission scenarios (e.g. assuming a complete transition to Euro V emission standard) to quantify the impact of the engine emission category on the mean emission factor and the additional contribution from shipping (see specific comments below). Text was added at the end of Sect. 3.3.1:

Overall, our dataset underlines the large emission reduction potential when modernizing the engines of inland vessels. In this study ~ 30 % of the fleet's motors were fulfilling the Euro IIIa standard, only ~ 0.1 % the recent Euro V standard. Assuming a future scenario, where 50 % of the ships have been equipped with a Euro IIIa stage motor and 50 % with a Euro V motor, would already halve the $NO_x$ and PM emissions based on our collected data. A complete transition to Euro V standard would even reduce $NO_x$ and PM emissions by more than 90 %. In particular, the use of DPF technology would also result in a drastic reduction of BC. Referring to Table 2, the additional $NO_2$ contribution from shipping at BRI and RIV could be reduced to 0.12 µg m$^{-3}$ respectively 0.04 µg m$^{-3}$, corresponding to less than 1 % of the total $NO_2$ immissions. Similarly, the PM contribution from shipping could be reduced to 0.04 µg m$^{-3}$ (0.5 %) at BRI, directly above the shipping lane. An accelerated modernization of the fleet would thus have a measurable impact on local air quality, especially when transferring the observations to more heavily frequented sections along the Lower Rhine (e.g. Duisburg).

We also added a statement in the conclusions section, stressing the large emission reduction potential of a modernized fleet.

Increasing the percentage of motors fulfilling the Euro IIIa and the Euro V standard to 50 % each, would already halve emissions and thus immissions in close vicinity to the shipping lane. A complete transition to Euro V has the potential to reduce them by ~ 90 % and to improve air quality in cities along inland rivers with heavier ship traffic and port activities than observed for Worms.

*Figure S1: Correlation plot of $CO_2$ peak areas derived from Licor and ICAD measurements at BRI.*

Units should be ppm*s

The units were changed to ppm*s.

On this occasion, we also corrected a few minor technical mistakes and imprecise expressions in the manuscript. These corrections do not affect the meaning of the text.

p.10, l.10: To be consistent with Eq. (1) we changed the nomenclature in Eq. (2) and (3) and also use $C$ for the peak area instead of $\Delta$.

To calculate the initial concentration at the time of emission (Eq. 2), the amount of $O_3$ that has been consumed by the reaction with NO (($C_{O3}$)$_{decrease}$) was subtracted from the measured $NO_2$ signal (($C_{NO2}$)$_{measured}$), resulting in a somewhat lower concentration of $NO_2$ at the time of emission (($C_{NO2}$)$_{initial}$) while $NO_x$ is conserved. The initial $NO_2$-to-$NO_x$ ratio is then defined by Eq. (3).

$$(C_{NO2})_{initial} = (C_{NO2})_{measured} - (C_{O3})_{decrease} \qquad \text{(Eq. 2)}$$

$$\left(\frac{NO2}{NOx}\right)_{initial} = \frac{(C_{NO2})_{initial}}{(C_{NOx})_{measured}} \qquad \text{(Eq. 3)}$$

p.10, l.14: the time of occurrence of the peak maximum

p.12, l.30: [...] at a larger distance of about 200–800 m […]

p.14, l.19: For the Elbe river in Wedel […]

p.22, Fig. 8b: Labels on x-axis changed to P1, P2, etc.

Additional literature to be considered:

Buffaloe, G. M.; Lack, D. A.; Williams, E. J.; Coffman, D.; Hayden, K. L.; Lerner, B. M.; Li, S.-M.; Nuaaman, I.; Massoli, P.; Onasch, T. B.; Quinn, P. K.; Cappa, C. D. Black Carbon Emissions from In-Use Ships: A California Regional Assessment. Atmos. Chem. Phys. 2014, 14, 1881−1896.

Gysel, N. R.; Russell, R. L.; Welch, W. A.; Cocker, D. R. Impact of Aftertreatment Technologies on the In-Use Gaseous and Particulate Matter Emissions from a Tugboat. Energy Fuels 2016, 30, 684−689.

Jiang, Y.; Yang, J.; Gagné, S.; Chan, T. W.; Thomson, K.; Fofie, E.; Cary, R. A.; Rutherford, D.; Comer, B.; Swanson, J.; Lin, Y.; Van Rooy, P.; Asa-Awuku, A.; Jung, H.; Barsanti, K.; Karavalakis, G.; Cocker, D.; Durbin, T. D.; Miller, J. W.; Johnson, K. C. Sources of Variance in BC Mass Measurements from a Small Marine Engine: Influence of the Instruments, Fuels and Loads. Atmos. Environ. 2018, 182, 128−137.

Nuszkowski, J.; Clark, N. N.; Spencer, T. K.; Carder, D. K.; Gautam, M.; Balon, T. H.; Moynihan, P. J. Atmospheric Emissions from a Passenger Ferry with Selective Catalytic Reduction. J. Air Waste Manage. Assoc. 2009, 59, 18−30.

Schlaerth, H.; Ko, J.; Sugrue, R.; Preble, C.; Ban-Weiss, G. Determining Black Carbon Emissions and Activity from In-Use Harbor Craft in Southern California. Atmos. Environ. 2021, 256, 118382

Sugrue R A, Preble C V, Tarplin A G and Kirchstetter T W 2022 In-use passenger vessel emission rates of black carbon and nitrogen oxides Environ. Sci. Technol. 56 7679–86

The suggested literature was added to the corresponding sections (see specific comments above).

---

## Author Comment (AC2)

**Reply to RC2**

We thank the referee for the review of our manuscript and the valuable comments and suggestions. In the following the referee's comments are repeated (in black) along with our replies (in blue) and changes made to the text in the revised manuscript (in red). Page and line numbers refer to those in the preprint version.

**General comments:**

The authors present the results of a comprehensive study on the differentiated assessment of inland shipping emissions on the Upper Rhine near Worms in Germany. They use a wide range of measurement techniques to detect gaseous ($CO_2$, $NO_x$ and $O_3$) and particulate (PNSD, PNC, $PM_x$, soot) air pollutants. Two sites have been selected for the measurements, allowing different scenarios to be mapped. One site was located on a bridge in order to record the plumes from passing ships close to the source. The second site was chosen directly on the banks of the Rhine. In this way, it is possible to determine the level of emissions that could affect people living near the Rhine. Particularly noteworthy is the methodology developed to identify individual ship plumes. The algorithm used avoids overlapping plumes, which can be caused by several ships passing at the same time. As a result, only clearly identifiable ship plumes are included in the evaluation. This results in a significantly reduced number of evaluable ship plumes and also reduces the number of individual ships in the composition of the shipping fleet. At the same time, the quality of the subsequent allocation and classification is significantly improved. In particular, the continuous long-term measurements over a period of one year provide a good picture of the emissions of the shipping fleet in this part of the Rhine. In addition, the emission factors can be calculated under real conditions, leading to a better understanding of the impact on inland navigation. This work represents a solid contribution and, in part, a new scientific approach to the measurement and characterisation of emissions from inland navigation under real conditions. The work is recommended for publication by this reviewer. The following suggestions may be incorporated into the authors' opinion.

We thank the referee for the positive evaluation of our manuscript and the helpful comments and suggestions below.

**Specific comments:**

*P3 L21*

*…high temporal resolution of ~1 s…*

Maybe one can mention, that the SMPS has a different and longer temporal resolution for a whole scan of the size range. Additionally, one could also explain the "problem" with scanning devices as a SMPS with a moderate sampling time. The assumption with a scanning device as the SMPS is that the aerosol spectrum does not change much over the time of a scan. However, this can occur with passing ships and short-term increases and thus lead to a distorted PNSD.

We agree that this is an important point to mention and added the following sentences to the text.

An essential part was the application of a fast mobility particle sizer (FMPS) measuring the whole size distribution in parallel by the use of numerous electrometers. For the detection of short-term increases like ship plumes it has a crucial advantage over commonly used scanning mobility particle sizers (SMPS) which need several tens of seconds for a scan and thus depend on the size spectrum not changing significantly over time.

*P4 L11*

*…Instrument-specific sampling lines of 4-5 m length…*

It seems that the calculated particle loss under 10 percent is relatively low. I would expect a higher particle penetration at this length of the sampling line. Did you use separeted sampling lines or did you use one sampling line with a higher volume flow and a manifold leading to the individual measuring devices?

We used separate sampling lines for each instrument at BRI but the flows for FMPS and AE33 were relatively high (10 lpm and 5 lpm respectively) so that diffusion losses for small particles could be limited. From Fig. S2 in the supplement it can be concluded that for the Grimm 11-D device (1.2 lpm flow rate) the overall loss rate for particles with diameter < 2.5 µm remains below 10 % but increases to more than 50 % at 10 µm. This is why we compared the measurement at BRI with the optical particle counter at RIV to make sure that the low particle count rate at diameters > 2.5 µm is a real feature of shipping emissions and not an artefact of the potentially high losses in the sampling line at BRI. We added the flow rates of the three devices to Fig. S2.

Calculated overall transmission losses with ~ 5 m sampling line for the instruments FMPS (flow rate 10 lpm), Grimm 11-D (flow rate 1.2 lpm) and AE33 (flow rate 5 lpm), derived using the Particle Loss Calculator Tool (von der Weiden et al., 2009).

*P4  L12*

*…to enable an undisturbed incoming flow.*

Doesn't the bridge itself generate turbulence that can contribute to influencing the wind field at the measurement site? Are downwind eddies possible that carry road traffic emissions down to the measurement site and superimpose the ship plumes as well?

We agree that the word 'undisturbed' is misleading. We tried to realize an incoming flow as undisturbed as possible within the technical possibilities at the bridge but it is likely that turbulence was generated close to the walls. Downwind eddies from road traffic are theoretically possible but the distance to the sampling line is > 5 m and the signal would clearly differ from the typical peak shape of ship plumes. Occasionally we observed a high atmospheric variability in the $NO_x$ signal (depending on meteorological conditions) which had to come from a local source close by, i.e. probably road traffic. But these periods were excluded in the analysis (see peak finding algorithm criteria) in order to avoid interference from non-ship sources. We modified the text accordingly.

Instrument-specific sampling lines of 4–5 m length were led through a hole in the wall, downwards to a point sharply below the edge of the bridge's base to reduce the impact of turbulence on the incoming flow. This way, the distance to the traffic lane on the bridge was also increased to ~6 m so that the impact of traffic emissions on the measurement site was minimized.

*P5 L5*

*…to avoid strong interferences from road traffic.*

You have chosen the locations to also avoid the influence of traffic related air pollutants. I am not familiar with the local conditions, but a look at the Nibelungen Bridge shows that this is a double bridge with two lanes each. What traffic volume can be expected there? Is there rush hour and congestion with traffic jams on the bridge? Especially with winds from northern directions, lee vortices could transport the TRAPs to the sampling point.

Both lanes of the bridge are highly frequented so there is the possibility of traffic jams and increased emissions during rush hour. As explained above, usually we did not notice any influence from road traffic on the bridge above. Occasionally we experienced an increased signal variability that can probably be attributed to road traffic (either emissions transported downwards via turbulent eddies or accumulated during the occurrence of thermal inversions) but this did not bias the analysis (see comment above).

Occasionally we observed an increased atmospheric variability in the $NO_x$ signal that probably originated from local non-ship sources like road traffic (favored in the presence of thermal inversions). These periods were, however, excluded from the analysis by defining appropriate criteria in the peak finding algorithm (see Sect. 2.3.1), so that the results were not biased.

*P6 Table 1*

Here the temporal resolution from the AIS signals is 1 s. To the best of my knowledge, an inland vessel sends a data set only every 10 s, depending on the current movement status.

Yes, like stated on p.8, l.6, AIS data is usually transmitted every 10 s. We changed the entry in Table 1 to 10 s.

*P12 L20-21*

*…further results […] refer to this instrument.*

This sentence is somewhat confusing, since in the coming chapters the results on RIV site will also be reported, which, however, were measured with the SMPS.

Our intention was to point out that - since the FMPS covered > 99 % of all particles detected in ship plumes - the results from the optical particle counter did not have a measurable impact on calculated emission factors and the additional contribution at

BRI. The SMPS measurement at RIV was only used to derive the background signal at RIV, whereas the additional contribution from shipping was estimated from the measured FMPS contribution at BRI (assuming a similar dilution factor as for $CO_2$). We slightly adapted the statement for clarification.

Since the FMPS (size range 6–520 nm) covered on average > 99.9 % of all particles detected in ship plumes, further results of PNC, $PM_1$ and $PM_{2.5}$ at BRI were based on this instrument.

*P12 L26-27*

The study by Pohl et al. was performed in Duesseldorf. So please change Upper to the Lower Rhine.

Thank you for noticing.

These results are consistent with a measurement study by Pohl et al. (2017) at the Lower Rhine.

*P14 L16*

*…as well as modern ships with exhaust after treatment…*

With regard to the CLINSH project. Weren't up to 40 ships retrofitted with downstream exhaust aftertreatment systems? Are the data or names of the ships available the authors to specifically read them out in their data set in order to be able to better scale up the positive effect of the emission reduction? This would be a good contribution, especially in view of the continuing increase in shipping traffic in the future.

Indeed, this would have been a good opportunity but from the best of our knowledge none of these ships passed the station in Worms during the measurement period. The majority of the ships retrofitted with exhaust aftertreatment systems in the CLINSH project are operated in the Netherlands or on the Lower Rhine in Germany. Nevertheless, we captured three ships in Worms fulfilling the recent Euro V standard (and thus using exhaust gas aftertreatment) which gave us the possibility to scale up the positive effect of an emission reduction, as also suggested by referee #1 (see our answer to referee #1).

In addition, on p.17, l.3 we now compare the mean emission factor of Euro V ships from this study with the mean emission factor and the emission reduction relative to CCNR II ships reported by the CLINSH project.

This low value and the ~ 90 % $NO_x$ reduction compared to CCNR II vessels agree very well with observations made by CLINSH (2022).

*P21 L10*

*With a BC fraction of 38 % for...*

It is (for me) not clear to which correlation the value is. Can you please more specify this. Is it $BC_{880}$ nm to total BC?

This value (measured at 880 nm by the AE33) refers to the fraction of BC mass relative to total PM derived from the FMPS measurement.

With a BC fraction (relative to total PM) of 38 % for upstream ships and 16 % for downstream ships our results indicate that a higher engine load leads to an increase in BC emission.

*P21 L14*

The proportion coming from biomass burning is mentioned here as about 10 % from biofuel combustion. Could it be a possible reason that the analyzed probe isn't just from ships because you also measure the background were also particles coming from wood fires, cigarette smoke, etc. Maybe there could be a hint, if the amount of bb is higher during the wintertime due to fireplaces?

We analyzed the biomass burning fraction at the time of occurrence of the peak maximum (average peak height ~ 50 $\mu$g m$^{-3}$) so that the background concentration (which was usually below 1 $\mu$g m$^{-3}$) can be considered negligible and a significant bias from non-ship sources can be excluded. We also found no difference between summer and winter season, indicative for residential heating.

The algorithm used by the AE33 to internally calculate the biomass burning fraction is based on multi-wavelength analysis (near-IR absorption relates to BC from combustion; a stronger absorption in the ultra-violet regime relates to organic material typical for biomass burning). Since the signature varies depending on combustion conditions and material, a clear separation can be difficult. Our statement on p.21, l.14 regarding biofuels is quite speculative but there are some hints in the literature that the fuel type has an impact on black / brown carbon measured by the aethalometer, e.g.:

Ma, M., Rivellini, L. H., Kasthuriarachchi, N., Zhu, Q., Zong, Y., Yu, W., Yang, W., Kraft, M., and Lee, A. K.: Effects of polyoxymethylene dimethyl ether (PODEn) blended fuel on diesel engine emission: Insight from soot-particle aerosol mass spectrometry and aethalometer measurements, Atmospheric Environment: X, 18, 100216, https://doi.org/10.1016/j.aeaoa.2023.100216, 2023.

We now write:

From multi-wavelength analysis of the AE33 (Sandradewi et al., 2008) we derived a mean fractional contribution from biomass burning of ~10 %, indicative for organic aerosol components that might be formed during the combustion of ship diesel blended with biofuel.

*P21 L25*

*…(see methods).*

Please refer to the chapter.

We now write (see Sect. 2.3.4).